# How children generalize novel nouns: An eye-tracking analysis of their generalization strategies

**Eleanor Stansbury**◉*, **Arnaud Witt, Patrick Bard**◉, **Jean-Pierre Thibaut**◉*

Laboratoire d'Étude de l'Apprentissage et du Développement, CNRS UMR 5022, Université de Bourgogne, Dijon, France

* Eleanor.Stansbury@u-bourgogne.fr (ES); jean-pierre.thibaut@u-bourgogne.fr (J-PT)

## Abstract

Recent research has shown that comparisons of multiple learning stimuli which are associated with the same novel noun favor taxonomic generalization of this noun. These findings contrast with single-stimulus learning in which children follow so-called lexical biases. However, little is known about the underlying search strategies. The present experiment provides an eye-tracking analysis of search strategies during novel word learning in a comparison design. We manipulated both the conceptual distance between the two learning items, i.e., children saw examples which were associated with a noun (e.g., the two learning items were either two bracelets in a "close" comparison condition or a bracelet and a watch in a "far" comparison condition), and the conceptual distance between the learning items and the taxonomically related items in the generalization options (e.g., the taxonomic generalization answer; a pendant, a near generalization item; versus a bow tie, a distant generalization item). We tested 5-, 6- and 8-year-old children's taxonomic (versus perceptual and thematic) generalization of novel names for objects. The search patterns showed that participants first focused on the learning items and then compared them with each of the possible choices. They also spent less time comparing the various options with one another; this search profile remained stable across age groups. Data also revealed that early comparisons, (i.e., reflecting alignment strategies) predicted generalization performance. We discuss four search strategies as well as the effect of age and conceptual distance on these strategies.

## Introduction

One remarkable feature of novel word learning is that it is achieved with a small number of learning stimuli and a small number of associations between the stimulus and the corresponding noun [1–3]. Understanding how children make sense of this paucity of information during novel word learning and which situations favor accurate word generalization is crucial for understanding language development. This problem has received considerable attention in recent decades and there are various competing theories regarding the explanatory factors [4–11].

**Funding:** This research was supported in part by a French ministry of Education PhD grant to Eleanor Stansbury, the French ministry of research for the grant ANR-18-CE28-0019-01 COMPARE to Jean-Pierre Thibaut, grants from the Conseil Regional de Bourgogne to Jean-Pierre Thibaut. The funders had no role in the study design, data collection and analysis, decision to publish, or preparation of the manuscript.

**Competing interests:** The authors have declared that no competing interests exist.

One prominent view of word learning and generalization is that a set of lexical constraints bias children towards particular dimensions of the stimuli [12–14]. Several lexical biases that minimize the number and type of dimensions children use as a generalization basis have been described in the literature [15–17]. For example, the shape bias leads children to generalize novel nouns towards targets with the same shape rather than the same color, size or texture [18, 19]. Given the large number of dimensions along which an object can *a priori* be described, constraining the number of potential dimensions novel words might refer to may help improve children's conceptually-based generalization.

Most authors would acknowledge the existence of the shape bias even though they may explain it in different ways, considering it as a bottom-up process [20] or as being susceptible to early conceptual influences (see [21], for an extensive discussion of the literature). Smith and colleagues [20] have proposed a four-step bottom-up model of how children learn to relate object names and attention to shape. At the end of this associative learning process, the object's shape becomes systematically associated with the object word. In Step 1, children associate names with individual objects (e.g., "this one is a ball"). In Step 2, first-order generalizations about individual object categories are made as children associate each category with a shape (e.g., balls are round). In Step 3, a higher-order generalization is made across learned categories and relates to the structure of object categories, which are labelled with a name—that is, these categories are defined by shape similarities. In Step 4, children have thus already learned to attend to shape when they learn novel object names and can rapidly learn novel names.

Even though various theories exist as to what makes a dimension relevant [22–29], a topic which is beyond the scope of the present project, the most important aspect of the process underpinning children's word generalization is that they come to pay attention to specific dimensions unifying the category (here, object categories). In this respect, to gain an insight into the processes that support novel noun generalization, it is necessary to investigate how children's attention changes during a novel noun generalization task.

Here, we concentrate on one novel noun learning situation that involves comparing several stimuli, in which lexical biases might be less appropriate or might guide the child towards irrelevant dimensions. For example, the shape bias would be irrelevant when shape is not relevant as a unifying cue [21, 30, 31].

## Novel noun generalization tasks

Novel noun generalization tasks are commonly used to investigate the way children extend novel nouns. In such tasks, children are usually shown a learning (otherwise known as a standard or reference) item. Past studies have made use of both pictures of objects and replications of objects. The item is associated with (often) a nonword (e.g. "*This is a buxi*" while showing a bracelet). The children are asked to choose another item among a set of options that could also have the same noun (e.g., "Which one is also a *buxi*?" among, for example, a necklace and a tyre in Fig 1A, below). The specific options from which the children must choose depend on what the researcher wishes to study. One answer is the taxonomic answer. In novel nous generalization tasks, the targeted answer is the taxonomically related answer because object nouns refer to items belonging to the same taxonomic category (e.g. another piece of jewelry in the present example). Numerous studies show that even young children know and follow this lexical bias towards taxonomic choice. The taxonomic choice is straightforward when taxonomy, shape and context converge. Here we concentrate on situations in which they are dissociated. We introduced distractors that were related to the learning items either thematically or perceptually but were not taxonomically related [19, 32–37].

It is important to stress that much of the evidence regarding novel noun learning and generalization has been obtained with a single learning item which is introduced together with its

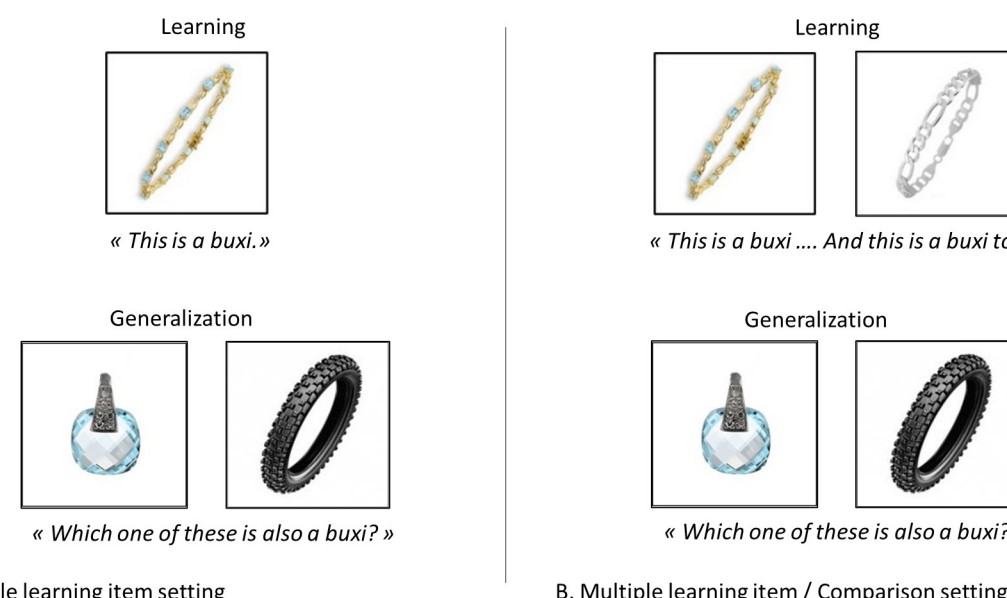

Learning

« This is a buxi.»

Generalization

« Which one of these is also a buxi? »

A. Single learning item setting

Learning

« This is a buxi …. And this is a buxi too. »

Generalization

« Which one of these is also a buxi? »

B. Multiple learning item / Comparison setting

**Fig 1.** Novel noun generalization tasks in A. a single learning item setting and B. a multiple learning item (comparison design).

noun. Then, the options are introduced with the learning item remaining in view. Recent studies have focused on variations of this learning task that might constrain children's learning of word meaning. In this respect, it has been shown that the possibility of comparing multiple learning items (e.g., two or more objects belonging to the same category such as two apples or two pieces of fruit) associated with a single noun, rather than a single learning item, favors taxonomic responding or the use of less salient unifying dimensions (see Fig 1B). To illustrate, in their seminal experiment, colleagues created a no-comparison and a comparison condition [32]. In the no-comparison condition, they followed the above design and presented 4-year-olds with one familiar object accompanied by a novel (made-up) name (e.g. *buxi* in Fig 1). They then showed the children two pictures, one of which was a taxonomic match and the other a perceptual match, and asked them which one had the "same name as the learning stimuli" (e.g. "*Which one is also a buxi*?"). The comparison condition was identical except that there were two learning stimuli, which were both presented with the same name. Results showed that most children selected the taxonomic match significantly more often in the comparison than in the no-comparison condition, in which the majority of children selected the perceptual match.

A large body of research has demonstrated the benefits of comparison designs for learning novel object names [38–40], adjectives [41], action verbs [42], relational nouns [43, 44] (see [45] for a meta-analysis; [46] for a synthesis). Considerable attention has also been paid to the conditions under which comparisons lead to better learning and generalization. For example, the effect of the number of items compared [37, 44], the perceptual similarity between the learning stimuli [47], or the conceptual distance between them [48, 49] have all been extensively studied.

According to colleagues, comparison conditions in novel-noun learning tasks elicit a deeper encoding of the compared learning items and lead children to progressively ignore irrelevant perceptual dimensions by emphasizing non-obvious conceptual properties [50, 51]. It is thought that this type of encoding is supported by processes that align features between

learning items. Alignment is an "umbrella" term which is widely used in analogical reasoning to describe how participants compare a base domain and a target domain in order to find both perceptual and higher-order relational similarities between domains [50]. Authors consider that, in the novel noun learning by comparison task, participants also align stimuli as they try to find commonalities, either perceptual or less obvious, deeper, ones [52–56]. Comparisons of multiple learning items at the beginning of a novel noun generalization task promote the alignment of their common features [32] and labels [51]. A more specific hypothesis is that the comparison process would start with an alignment of the most salient perceptual features. These earlier alignments would promote new comparisons that then lead to the discovery and alignment of less salient conceptual commonalities. Thus, the comparison process highlights conceptual features that would have remained unnoticed in a no-comparison situation; and the common representation that is extracted from this alignment process between item features is more conceptually based than the one that would have been built in a no-comparison learning condition. The fact that comparison situations lead to more conceptually-based (taxonomic in the case of nouns) choices when a noun is associated (rather than simply compared) suggests that nouns also contribute to the alignment process, most likely by encouraging subjects to engage in the process. It is most likely that the first step in the comparison process is driven by perceptual similarities [20, 25]. However, whether or not later stages of the comparisons, i.e., those that lead to the discovery of more conceptual features (e.g., both are fruits even though they are dissimilar), are driven entirely by these early perceptual associations or rely on higher, independent, conceptually-based inferences remains a topic of considerable debate. It could also be that later, conceptually-based representations of the stimuli (i.e. both belong to the same taxonomic category) are deeply rooted in early perceptual activations or, on the other hand, that there is a conceptual gap between the perceptual activations and deeper conceptual activations. This is a most important issue and one which has not as yet been definitively resolved [54–56].

## Steps and strategies in learning by comparison

The present experiment examines the way comparison situations lead to novel noun generalization in cases where novel words are used with familiar stimuli. Given existing lexical biases, we tackle the question of whether comparisons will lead to conceptually-based decisions, here a taxonomically-related item (e.g. a piece of jewelry in Fig 1), when a perceptual distractor (e.g. a tyre in Fig 1) and a thematically-related distractor (e.g. a hand) are available. Our main question is what visual search strategy, if any, do children follow to reach a conceptually based decision in this comparison design? We will examine this question using eye movement analyses. To date, the available research has focused on the benefits of comparison and on factors that might modulate its effects. To the best of our knowledge, the strategies that children adopt to make sense of the stimuli in a comparison design and generalize a novel object noun have not yet been explored. In our study we collect eye-tracking data allowing us to reveal the temporal organization of these strategies. Indeed, eye movements provide a precise reflection of the interactions between cognitive processes and the external visual stimuli [57–59] and can be used to reveal the profile of the search and generalization strategy.

In the following section, we borrow our candidate strategies for the lexical comparison task from the analogical reasoning literature. Both tasks are generalization tasks in which participants have to generalize properties from a base domain to a target domain. Even though the two tasks are of a different level of complexity, both are alignment tasks between domains [60–62]. In analogy tasks, the setting shows two items–A and B–(the source domain) which are connected by a specific relation. Then, the target domain is introduced with an item–C -, and

children have to select an item–D–that goes with C in the same way B goes with A. At a general level, both our noun generalization task and the analogical reasoning task request (1) the analysis of the source items (i.e., learning items and -A B- items) in order to extract commonalities; (2) the generalization of the information they have found to new items (i.e. one item among multiple options in both tasks).

One candidate strategy is known as the *Elimination strategy* and involves a one-by-one comparison of the items in the target domain with the items in the source domain, thereby eliminating inappropriate target items until one is found that the participant considers to be a solution [63]. A second strategy is the *Construction strategy*, in which participants start by studying the base pair composing the source domain (A and B, i.e. the two learning items) and search for a possible dimension that unifies it. They then apply this dimension in the target domain until they find a potential stimulus sharing the same dimension with A and B [64–66]. The two strategies can be summarized as follows: Elimination strategy: participants compare learning items and then compare learning items with each available generalization option; Construction strategy: participants compare learning items before applying what they have found to the target domain.

A third strategy has been described by colleagues, the *Task goal-directed strategy* [67]. This strategy refers to the fact that, at the beginning of the trial, younger children look very early in the trial at the set of options from which they have to find the solution (target domain) much more often than older children and adults. Thus, they do NOT systematically compare the A and B items in the source domain as in the construction strategy and nor do they eliminate the options one-by-one by comparing them to A and B. The authors interpreted this pattern as reflecting children's difficulty in inhibiting the main goal of the task, which is "finding the one that goes with C". Following this line of reasoning, strategies across age groups may reveal children's increasing ability to focus on the tasks source of information (i.e., the learning items) before searching for the task solution. If this is the case, our main interest is to assess whether children who manage to inhibit this early search within the solution set are also those that succeed the best.

## The present study. Search strategies in novel word learning and generalization

The present study seeks to establish the strategies that participants adopt to reach a taxonomic generalization decision when a comparison design is used in a novel noun generalization task. These strategies will be captured in the form of the temporal organization of gazes up to the point at which children select one stimulus as a referent for the novel noun. To identify the adopted strategies, children have to integrate information coming from various stimuli when the distractors are more salient or as salient as the taxonomic choice. It is therefore important to understand how children organize their searches to reach a decision, that is the temporal dynamics of the novel noun generalization strategies.

The five main aims were to (1) provide a description of novel noun generalization strategies in our comparison design, in terms of patterns of gazes and switches during the trial; (2) interpret this data in terms of a set of strategies; (3) assess the organization of the search and how it interacts with age and task difficulty (i.e., conceptual distance between items); (4) assess whether early gazes predict generalization performance; (5) provide a description of the strategies adopted by good and poor generalization participants.

We tested 5-, 6-, and 8-year-old children in the novel noun generalization task. Six- and eight-year-olds were our two main target groups as we knew [37] that the younger group might still have difficulties with the task, to be attracted to salient distractors available among

the generalization options alongside the taxonomic option, despite the fact they can categorize when less distracting options are introduced [68]. At the same time, we also knew that the older group would primarily give taxonomic answers. Thus, eight-year-olds were expected to produce a majority of mature strategies, whereas six-year-olds were expected to produce strategies that would reflect a greater influence of the distractors. We also added a five-year-old group after pretests showing that this age group might yield reliable eye-tracking data (see below) and would also testify to young children's difficulties in conceptualizing taxonomic relations. Indeed, previous studies, revealed difficulties to collect extensive eye-tracking data involving many trials at such a young age and that the data contains a lot of noise (in the case of analogies, five-year olds revealed a higher percentage of lost data [67].

We also manipulated the level of difficulty in terms of the conceptual distance between either the learning items themselves or between the learning and generalization options, as conceptual distance might influence the strategy children follow (see below). Indeed, colleagues [49] showed that all conceptual distance conditions differ in difficulty given that it should be easier to find unifying features for conceptually close objects than for distant objects. Here, conceptual distance refers to the taxonomic distance in terms of categorization levels [69]. Following previous studies [39], we define distance in terms of steps in the taxonomic hierarchy: two apples belong to the same basic category and are at distance 0. An apple and an orange belong to the same immediate superordinate category and are at distance 1. Finally, an apple and a piece of meat are at distance 2 as their common level of categorization (food) is separated from them by an intermediate superordinate level of categorization (fruits and meat). In previous work [39], colleagues manipulated the conceptual distance between the learning items themselves (e.g., two bracelets versus a bracelet and a watch) and between the learning items and the generalization items (or options) (e.g., a piece of jewelry, near condition, versus a bow tie, distant condition). Results revealed fewer taxonomic choices in the distant generalization condition than in the near generalization condition. Taxonomic generalization at different levels of categorization has also been studied using a single-stimulus design and it has been found that the basic level is the spontaneous naming level rather than the superordinate or subordinate levels of categorization [14, 33]. Overall, we investigate how potential search strategies interact with age and conceptual distance.

In order to reveal these search strategies, we will focus on switches between areas of interest (AOIs) and fixation times on each AOI. Switches reveal comparisons between items and thus the alignment between their features [70]. Fixations reveal which items have been gazed at the most and are thus considered the best sources of information.

Hence, the main goal of the analyses is, first, to provide a description of the temporal dynamic of the novel noun generalization strategies. How do children reach a solution? By comparing the learning items before they start looking at the options? Without these early comparisons? Do they compare the options themselves or not? Are they mesmerized by the shape-similar option? Do strategies change with age? How would these questions translate in terms of the above strategies? An Elimination strategy would involve first comparing/aligning the learning items and then going on to compare each of the available generalization options with the learning items, thereby progressively eliminating the generalization items until only one item is left. This should be seen initially in the form of switches between learning items (i.e., comparison of learning items), followed by switches between each of the generalization options and at least one learning item (i.e., search by elimination). The elimination strategy may also be compatible with an even distribution of gazes between the different stimuli, i.e., both the learning items and the options, throughout the entire trial.

A Construction strategy would involve comparing the learning items (characterized by significant switches between learning items). This strategy can be interpreted in terms of

alignment and the identification of common features. The participants should then compare each generalization item with each of the others until one item is chosen as the generalization solution. This strategy will first involve longer periods spent looking at the learning items before the participants then start gazing towards the available generalization options. The Construction strategy predicts very few early gazes towards the options, whereas the Elimination strategy is compatible with early gazes towards the solution set.

These strategies differ mainly in the way the solution set is explored. The Elimination strategy predicts a high proportion of gazes from learning stimuli towards options and very few comparisons between them, whereas the reverse is true for the Construction strategy. The third strategy, namely the Task Goal-directed strategy [67], predicts more switches to and/or longer gazes at the options early in the trial. One way to reveal it is to examine the ratio of switches between the two learning items to switches between either of the learning items and any of the options. Low values on this index at the beginning of the trial should reveal early explorations of the options, that is around the stimuli that are related to the main goal of the task, which is to select one option [44]. We predict an age-related increase in the index value.

Fourth and finally, the associationist views on the importance of shape could lead to an additional strategy in which children, and in particular younger children, first explore the perceptually similar option. This view (Salient Shape view) predicts that the length of the first gazes towards the perceptual distractor should be greater than that of early gazes towards other options or that the number of early switches between learning items and the perceptual distractor should be greater than that of early switches between learning items and other options.

With regard to the role of conceptual distance, we hypothesize that this distance might modulate not only children's generalization performance [44, 49] but also their *strategies*. For example, colleagues [63] demonstrated that children shift from one strategy to another as a function of task difficulty, relying on the elimination strategy when difficulty increases, presumably because the elimination strategy is less costly. Accordingly, younger children should adopt the elimination strategy more often than older children in the most difficult conceptual distance condition. The reason for this is that the constructive matching strategy is cognitively more demanding since it involves storing the source item's representation in memory and using it to choose among multiple options without referring back to the source items.

Finally, we will analyze the distribution of early gazes and its relation with the taxonomic decision at the end of the trial and assess whether different patterns can predict which answer a child will choose. Indeed, as shown by analogy tasks [67], the pattern at the beginning of a trial has predictive power. The authors showed that the distribution of gazes in the first third (Slice 1) of a trial predicted whether the trial's outcome would be a correct (i.e. taxonomic) generalization or an error. Hence, we predict that the number of early gazes towards the learning items will increase with age and that a larger number of these gazes will be correlated with a larger number of taxonomic generalizations. In this perspective, we will analyze whether early gaze profiles distinguish between high- and low-achieving participants. We hypothesized that taxonomic choices require a careful analysis of the learning items and fewer early gazes towards the options, meaning that high achievers would have a higher ratio than low achievers of early gazes towards the learning items compared to gazes towards options.

## Methods

### Participants

Two hundred and thirty-six children subdivided into three age groups were tested individually in a quiet room at their school. Seventy 5-year-old children were recruited (mean age = 63 months; range: 58–69), together with ninety-six 6-year-old children (mean age = 80 months;

range: 65–92), and seventy 8-year-old children (mean age = 100 months; range: 91–116). Informed consent was obtained from their school and their parents.

A priori power analyses were conducted using G*power [71] for sample size estimation. For generalization performance data the power analysis was based on data from a previous study [39]. They analyzed the effect of learning distance and generalization distance on children's percentage of taxonomic generalizations. Learning distance's effect size was .39 and generalization distance's effect size was .50. With a significance criterion of α = .05 and power = .80, the minimum sample sizes needed with these effect sizes were respectively n = 56 and n = 12. For eye-tracking data the power analysis was based on data from colleagues [67] who analyzed proportion of fixation times and number of saccades as a function of age, time slice and item or saccade type. Interaction' effect size between age * time slice * item and between age * time slice * saccade were respectively .39 .42. With a significance criterion of α = .05 and power = .80, the minimum sample sizes needed with these effect sizes were n = 12 and n = 8. Accounting for a potential attrition rate with eye tracking data of 20% based on previous research [39] and a considering the large sample size estimation [56] at least an additional (20% of 56) 11 participants needed to be recruited, for a total sample size of 67 participants per age group. Thus, the minimal obtained sample size per age group (N = 70) is adequate to test the study hypothesis.

The procedure complied with the declaration of Helsinki (1964) and was ethically reviewed and approved within the framework of an official agreement (convention no.: 2019–0679 and endorsement no.: 2020–0566) between the French National Education Ministry's Academic Inspectorate ("Inspection Académique de Côte d'Or"), the University of Bourgogne, and our laboratory.

## Materials

We used fourteen familiar categories which were adapted from Thibaut and Witt (2017) (see S1 Table for a full list of materials). An experimental set of seven pictures was built for each category (see Fig 2). This comprised a standard learning item belonging to the targeted category (Fig 2, $L_1$, e.g. a bracelet), and two other learning items, one conceptually close to the standard learning item (from the same basic level category; Fig 2, $L_2c$, e.g. a curb chain) and one conceptually far from it (from the same superordinate category, Fig 2, $L_2f$, e.g. a watch). All learning items ($L_1$, $L_2c$ and $L_2f$) were perceptually similar.

The experimental set also included a taxonomically near generalization item (Fig 2, Ta Near; e.g. a jewel pendant) from the same immediate superordinate-level category as the standard learning item and a distant generalization item (Fig 2, Ta Distant; e.g. a bow tie) from a more remote superordinate category level than that of the standard learning item. Either the near or the distant option was shown during an experimental trial.

Finally, two distractors were also included in the set. One was perceptually similar to the learning items but belonged to a semantically unrelated category (Fig 2, P, e.g., a tire). The other was thematically related to the learning objects but not taxonomically or perceptually related to them (Fig 2, Th, e.g., a hand) (See S1 Table. A for a full list of materials).

Items were familiar implying that children already know the item's label in their own language. How much this prior knowledge about the items affects children's results may be questioned. First, if this were the case children's generalization performances would be correlated to their vocabulary levels, which appeared not to be the case. Second, children's sensitivity to the distractors indicates that their novel word extensions were not entirely driven by their prior knowledge. Third, recall that in a previous study [39] colleagues used the same materials but including a no-comparison condition which led to a majority of perceptual distractors. If

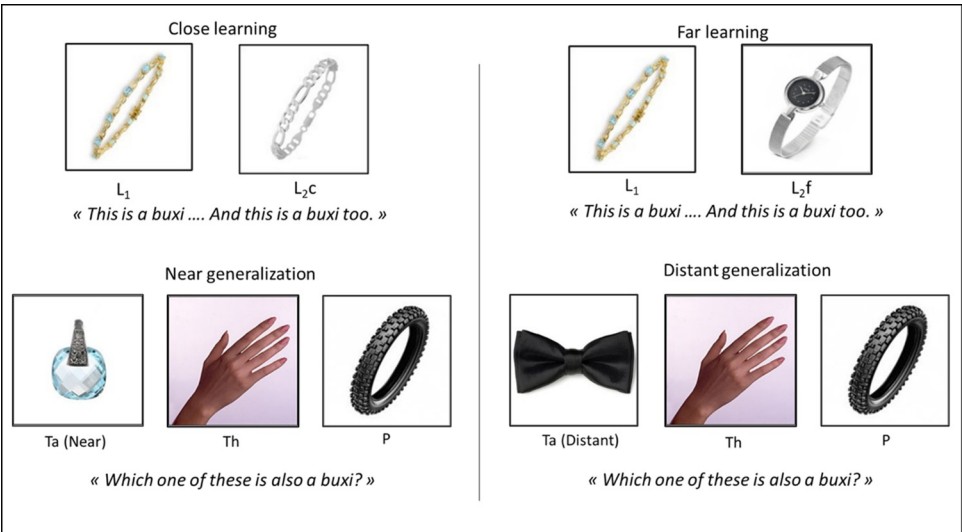

**Fig 2. Example of a stimulus set and instructions adapted for experimental conditions resulting from crossing learning distance (close vs. far learning) and generalization distance (near vs. distant).**

children were relying on their language knowledge, they should have massively selected the taxonomic choice.

Items were 2D pictures rather than 3D objects because 2D pictures have most often been used in novel noun generalization studies and, given that we already know [39] that our objects are realistic representations of the corresponding stimuli. In addition, they are well suited for studying children's eye movements, since their location is easy to control. Even though, it might be thought that 2D pictures are not as informative as 3D items in terms of perceptual input, our question was whether the perceptual distractor would attract sufficient attention, especially in the case of familiar stimuli. As mentioned above, the same stimuli were used in a no-comparison condition and found that children selected the perceptual match above chance level, as predicted by a shape bias.

Independent similarity ratings from 54 university students (mainly psychology students, who participated in return for course credits, mean age = 20.15 years; 47 females) were recorded to control the material. We first controlled for the differences in the conceptual distance between conceptually related items. Conceptual similarity ratings confirmed that the close learning stimuli were conceptually closer to their related standard learning stimulus than the far learning stimuli, (scale 1–7, $M_{Close}$ = 5.52, 95% CI [5.32, 5.72]; $M_{Far}$ = 4.44, 95% CI [3.89, 4.99]), $t(13)$ = 3.31, $p < .006$, $d = 0.89$ (Bonferroni-corrected p-value threshold .05/8 = .00625), and that near generalization stimuli were conceptually nearer the learning stimuli than distant generalization stimuli, (scale 1–7, $M_{Near}$ = 3.75, 95% CI [3.45, 4.05]; $M_{Distant}$ = 2.48, 95% CI [2.06, 2.90]), $t(13)$ = 4.41, $p < .006$, $d = 1.81$.

The taxonomically related generalization options have to be perceptually less similar to the learning items than the perceptual distractors. Perceptual similarity ratings revealed that the perceptual choices were perceptually more similar to the learning items than the taxonomic choices in both the close and the far learning conditions (scale 1–7, $M_{Perceptual}$ = 4.77, 95% CI [4.49, 5.05]; $M_{Near}$ = 2.13, 95% CI [1.81, 2.45]; $M_{Distant}$ = 1.86, 95% CI [1.550, 2.21]), $t(13)$ = 10.59, $p < .001$, $d = 2.83$, and $t(13)$ = 11.63, $p < .001$, $d = 3.11$.

We also tested whether the thematically related items were semantically related to the learning items in both close and far learning conditions (scale 1–10, $M_{semantic}$ = 7.74, 95% CI [7.34,

8.14]; $M_{Near}$ = 6.30, 95% CI [5.38, 7.22]; $M_{Distant}$ = 4.08, 95% CI [3.07, 5.09]), $t(13)$ = 2.95, $p <$ .05, $d$ = 0.79, and $t(13)$ = 6.62, $p <$ .001, $d$ = 1.77.

Importantly, we also performed perceptual similarity and conceptual similarity ratings between the close learning stimuli (e.g., two apples) and the far learning stimuli (e.g., an apple and a cherry), on the one hand, and the taxonomically related generalization item, on the other. They showed that, overall, the generalization stimuli were equally distant, both perceptually and semantically, from both types of learning item. This was true for both types of generalization items. Near generalization items were perceptually as, similar to the close learning items (scale 1–7, $M_{Close}$ = 2.18, 95% CI [1.82, 2.54]) as to the far learning item ($M_{Far}$ = 2.02, 95% CI [1.71, 2.33]). This was confirmed by a Student t-test ($t(13)$ = 1.58, $p$ = .14, $d$ = 0.42). Near generalization items were also as conceptually near the close learning items (scale 1–7, $M_{Close}$ = 3.09, 95% CI [2.82, 3.36]) as the far learning items ($M_{Far}$ = 3.36, 95% CI [3.05, 3.67]), as confirmed by a Student t-test ($t(13)$ = -2.96, $p$ = .01, $d$ = -0.80; Bonferroni-corrected p-value threshold .05/8 = .006). Distant generalization items were as perceptually similar to the near learning items (scale 1–7, $M_{Close}$ = 1.88, 95% CI [1.56, 2.20]) as to the far learning items ($M_{Far}$ = 1.86, 95% CI [1.47, 2.25]), as confirmed by a Student t-test ($t(13)$ = .24, $p$ = .81, $d$ = 0.06). Distant generalization items were also as conceptually distante from the close learning items (scale 1–7, $M_{Close}$ = 2.18, 95% CI [1.82, 2.54]) than as the far learning items ($M_{Far}$ = 2.02, 95%CI [1.71, 2.34]), as confirmed by a student t-test ($t(13)$ = -.19, $p$ = .85, $d$ = -0.05). This was crucial because we wanted to avoid the possibility that performance differences between near and distant generalization items might be due to perceptual or semantic similarity differences between them and the learning items. For example, if we observed a difference between near and distant generalization items (e.g. between jewelry pendant and bow tie), we did not want it to be due to semantic information (e.g., the fact that the pendant is more thematically related to the bracelet than the bow tie) other than the difference in conceptual distance (See S2 Table for details of the ratings).

Fourteen different bisyllabic labels (pseudo-words) [72], which are easier to remember than monosyllabic pseudo-words, were used to name the learning objects in each trial. Syllables were of the CV type, which is the dominant word structure in French (from Lexique.org, bari, buxi, daxo, jito, malan, missi, muno, nati, peco, pina, rula, sefu, soki, vira, youma, zatu [73]).

Two learning items were used in each trial, either the standard learning item and the close learning item ($L_1 – L_2c$, close learning condition) or the standard learning item and the far learning item ($L_1 – L_2f$, far learning condition). One of the two generalization items was used (near or distant item according to the generalization condition) together with the perceptually and the thematically related distractors. The two learning items were displayed side-by-side at the top of the screen and the three generalization options were displayed side-by-side below the learning items (see Fig 2). The learning items' positions were randomized, as were the positions of the generalization items. For each trial, the pictures were displayed simultaneously on a Tobii T120 eye-tracker device with a 1024x768 screen resolution until the answer was given.

Each picture constituted an Area of Interest (AOI) for the eye-tracking analysis. Picture size was 500x500pixels and each picture was surrounded by a square black outline. The boundaries of the AOIs corresponded to a picture's outline. A standard fixation cross was shown for 3 seconds between each trial. Each experimental session started with a standard calibration phase conducted after the three warm-up trials. The experiment was run with E-prime® software (Psychology Software Tools, Pittsburgh, PA).

Children's general knowledge and language development were tested using the EVIP (Echelle du Vocabulaire en Images de Peabody) standardized French vocabulary test. As expected, children's scores improved with age, but no significant correlations were found between vocabulary scores and generalization scores (Pearson partial correlation r = 0.07, $p$ = 0.45).

## Procedure

The child and the experimenter were seated at a table in a quiet room in the child's school. The experimenter introduced the experiment as a game to be played with a bear named Sammy. "Hello, we are going to play together, and we are going to play with a bear called Sammy. Look, this is Sammy, he lives far away from here and speaks a different language; he does not speak like us and we are going to learn his language."

It was critical to limit external elements that might have influenced the children's gaze patterns during the task (e.g., hand movements by the experimenter, sequential item appearance on the screen etc.). To eliminate the effect of the item's appearance, all of the trial's stimuli always appeared simultaneously on the screen, thus making stimulus location highly predictable (learning and option items). When the items appeared, the experimenter gave the trial instruction: "*See, Sammy's mummy* says *this is a buxi. And this is a buxi too. Sammy must find another buxi. Can you show which one of these three is also a buxi, to help Sammy? Can you point to the other buxi?*" To eliminate any effect of the experimenter's movements, he did not make any hand movements towards the screen during testing. To ensure that this did not affect children's understanding of the task, a warm-up phase, intended to explain the task to the children, was conducted before eye-tracking recording started.

In the warm-up phase, children saw three trials that were identical to those seen later during the test phase except for the instruction given. In the first warm-up trial, the experimenter added location information and pointing movements to the items he was talking about. "See, Sammy's mummy says this item at the top of the screen is a buxi. And this one next to it is a buxi too (Exp points to them). Sammy must find another buxi. Can you show which one is also a buxi, to help Sammy? Can you point to the other buxi among these items at the bottom of the screen (pointing at the three options on the screen)?" In the second warm-up trial, the experimenter gave the same instruction without pointing and without location information (e.g., "at the top of the screen"). In the third warm-up trial, the experimenter gave the same instruction as in the test trials, still without pointing. If the child did not answer in the second warm-up trial, the experimenter repeated the instruction together with the pointing movements. It was decided that children who did not answer in the third warm-up trial would be excluded because we could not be certain that they had understood the task. However, no child was excluded at this stage. After the warm-up trials, the experimenter started the eye-tracking calibration phase, which was followed by the experimental trials themselves.

During the test phase, the experimenter showed the child 14 trials as described above. The experimenter used no hand movements or other ways of highlighting the relevant items. The child chose an item by pointing to it and the experimenter selected it with the mouse. Response times were recorded from the time when the item appeared on screen to the experimenter's click, which also ended the trial and cleared the display.

The order of the fourteen experimental trials was randomized, as were the names used. A blank screen with a central fixation cross was shown for 3 seconds between each trial. Participants knew the items taken from Thibaut & Witt (2017; 2023), who tested 4- and 5-year-old children with these items.

Eye-tracking data were recorded as follows. The Tobii 120 eye-tracking device, equipped with two cameras, recorded the position of the two eyes every 8.33ms. Raw data consisted of all spatial coordinates of eye positions throughout trials [74].

## Results

### Performance data

Performance was measured as the percentage of taxonomic generalizations (i.e., percentage of taxonomic choices) and children's reaction times in milliseconds. We first examined the

strategies that lead to taxonomic generalization. Following standard practice, we initially focused on correct (i.e., taxonomic answers) trials only, because we wanted to describe search strategies that lead to a correct answer. It is more difficult to interpret incorrect answers as they may result from a real choice or from random choices when children are inattentive (see below for a comparison of high-achieving and low-achieving participants). We also compare the strategies followed for errors and correct answers (see below)

**Generalization scores.** First, we analyzed the percentages of taxonomic choices. We ran a three-way repeated-measures ANOVA on participants' percentage of correct answers, with age (5, 6, 8 years old) and learning distance (close, far) as between-factors and generalization distance (near, distant) as a within-factor. The analysis revealed an effect of generalization distance: $F(1, 179) = 54.74, p < .001, \eta_P^2 = .23$, and a main effect of age: $F(2, 179) = 9.95, p < .001, \eta_P^2 = 0.1$ and an interaction between generalization distance and age: $F(2, 179) = 3.30, p = .04, \eta_P^2 = .04$ (see Fig 3). All other main effects and interactions failed to reach significance ($p < .05$).

For 6- and 8-year-old children, an a posteriori Tukey analysis revealed that near generalization scores were significantly higher than distant generalization scores ($p_s < .001$). There was no significant difference between near and distant scores for 5-year-old children. One-sample T-tests revealed that the three age groups scored above chance (set at 33%), except for 5-year-old children, who scored at chance in the distant generalization condition. Indeed, 8-year-old children scored above chance in both near and distant generalization conditions: near generalization, $t(184) = 10.30, p < .001$; distant generalization: $t(184) = 5.71, p < .001$. Six-year-old children were also above chance in both generalization conditions: near generalization, $t(184) = 8.50, p < .001$; distant generalization, $t(184) = 5.34, p < .001$. Five- year-old children scored above chance in near generalization conditions: $t(184) = 3.58, p < .001$, but at chance in distant generalization conditions $t = 1.43, p = .16$.

**Reaction times.** We ran a three-way repeated-measures ANOVA on children's average reaction times for trials that led to correct answers, with age (5, 6, 8 years old) and learning

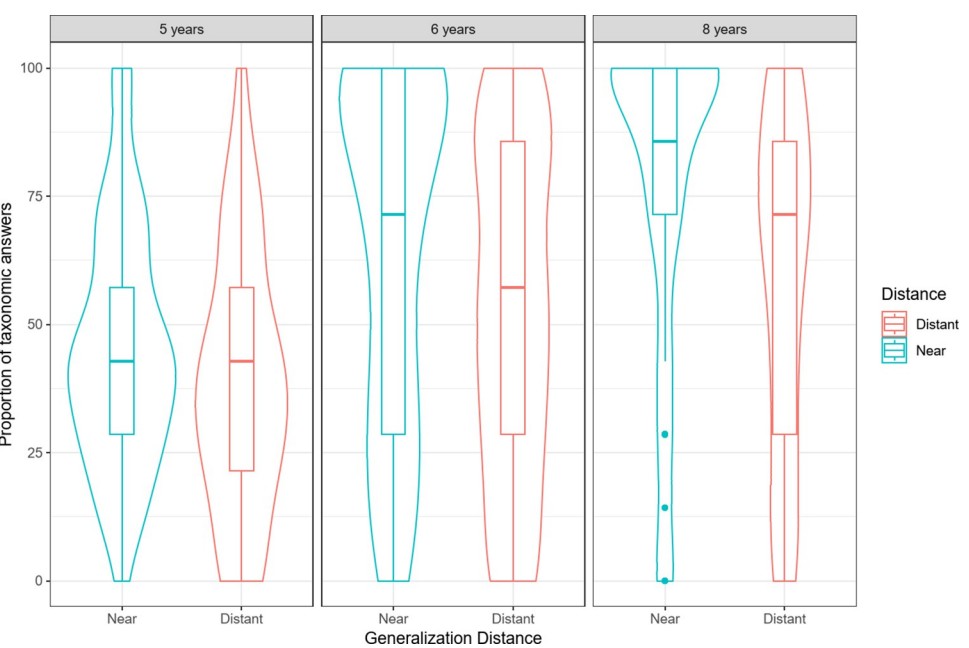

**Fig 3. Percentage of taxonomic choices as a function of age (5, 6 and 8 years old) and generalization distance (near, distant).**

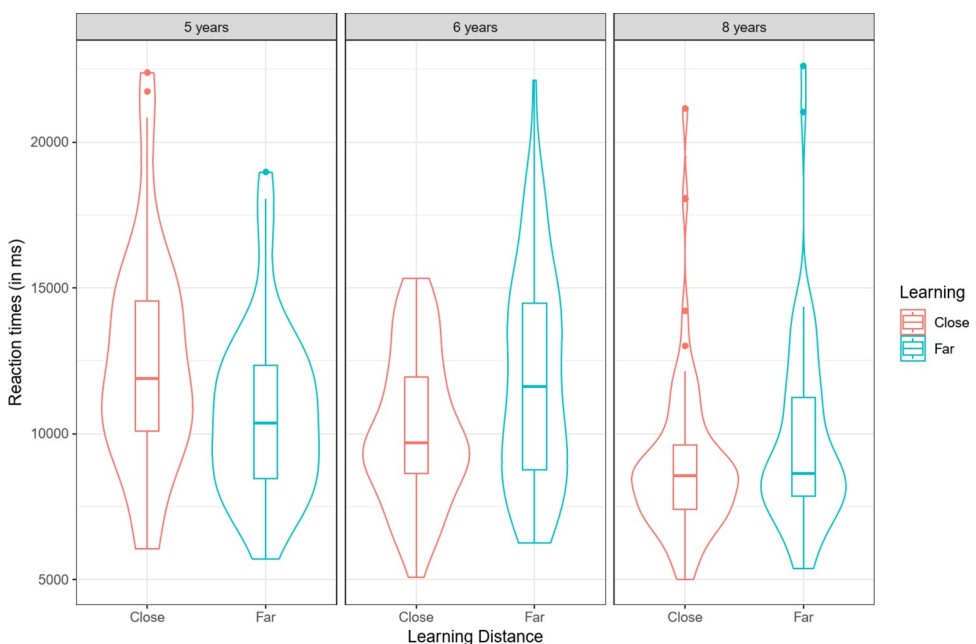

**Fig 4. Reaction times (ms) for correct answers as a function of learning distance (close, far) and age (5-, 6-, 8-years-old).**

distance (close, far) as between-factors and generalization (near, distant) as a within-factor. The analysis revealed an effect of age, $F(2, 182) = 7.87$, $p < .001$, $\eta_P^2 = .08$, and interactions between learning distance and age, $F(2, 179) = 5.38$, $p < .01$, $\eta_P^2 = .06$ (see Fig 4), and between generalization distance and age, $F(2, 182) = 3.42$, $p = .04$, $\eta_P^2 = .04$ (see Fig 5). Five-year-olds were slightly faster in the distant generalization condition, but performed at chance. Therefore, this might mean they failed to perform the task in this condition. An a posteriori Tukey analysis revealed that 6-year-old children's reaction times were significantly faster in close than in far learning trials ($M_{close}$ = 10,208 ms, $M_{far}$ = 12,307 ms; $p = .046$). An a posteriori Tukey analysis revealed that reaction times decreased with age between 8 and 6 years of age (5- vs 6-year old children: $p = .84$; 6- vs 8-year old children: $p = .002$), showing that children find the task easier with age.

Taken together, these results show a consistent pattern of results in which the distant generalization condition was the most difficult condition, in terms of both RTs and the percentage of taxonomic choices, thus confirming recent performance data reported by Thibaut and Witt (2023).

### Eye-tracking data: Revealing children's search strategies

To meet the normality requirements for ANOVAs, we applied normal transformations to our data. Log transformations were applied to the number of switches. We applied Greenhouse-Geisser or Huynh-Feldt corrections to F scores when sphericity scores required them.

**Collection and processing of eye-tracking data.** Data from seven children were removed because eye-tracking calibration was unsuccessful (one 5 year-old -, four 6 year-old—and two 8-year-old children) due to the fact that inadequate contrast between eye color, room lighting and screen lighting made the pupil too difficult to detect. Data from a further twelve children was incomplete because they interrupted the task or because eye-tracking calibration was interrupted during the task because the children were unable to stay still (ten 5 year-old and two 6 year-old children). This data set was also excluded from further analyses.

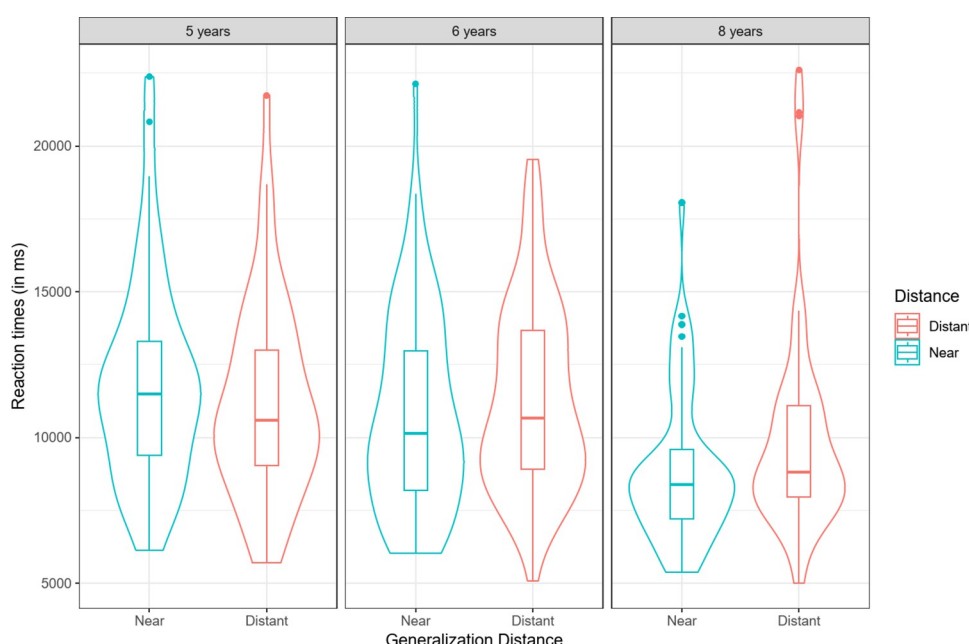

**Fig 5. Reaction times (in ms) for correct answers as a function of age (5, 6, 8-year-old) and generalization distance (near, distant).**

We also excluded trials for which more than 50% of eye positions had been lost and we excluded subjects for whom more than 50% of trials had been excluded. This resulted in the exclusion of 32 subjects (twelve 5-year-old, ten 6-year-old; and ten 8-year-old children). Among the remaining participants, we lost an average of 1.7 trials for 5-year-old children, 1.1 trials for 6-year-old children and 1.1 trials for 8-year-old children.

Eye-tracking data was divided into three equal time slices. The length of a time slice was equal to a third of a trial's total fixation time. We first focus on gazes towards AOIs and switches (transitions). Gaze duration (or looking times) for areas of interest (i.e., AOIs) tells us which stimuli were attended to and for how long while solving the problem and also reveal the depth of processing of the item. Switches between items (or saccades or transitions) tell us which items were compared and can be interpreted as an attempt to find commonalities between X and Y [67]. The two measures reveal different aspects of the search, as a participant could study an item for a large amount of time without comparing it with other items, while gaze duration, unlike transitions, tells us nothing about which items were compared during trials.

**Learning distance.** We first performed a five-way repeated-measures ANOVA on the proportion of fixation times or the log-transformed number of switches (see below), with learning distance (close, far) and age (5, 6, 8) as between-subject factors and generalization distance (near, distant), time slice (beginning, middle, end) and AOI (L, Th, Ta, P) or switch type (LL, LTh, LTa, LP, ThTaP) as within-subject factors. (For full results, see S1 Data). In the case of proportions of fixation times, learning distance was involved in only two significant interactions, learning distance, time slice and AOI $F(6, 714) = 2.32$, $p = .02$, $\eta_P^2 = .02$ and learning distance, generalization distance, time slice, AOI $F(6, 714) = 2.12$, $p = .049$, $\eta_P^2 = .01$. However, these interaction's effect sizes were small and post hoc tests did not reveal any differences between close and far learning groups (see S1 Data for all $p$-values). In fact, these interactions resulted from small differences between AOIs across time slices.

In the case of the log of the number of switches learning distance was only involved in the interactions between learning distance and time slice, $F(2, 420) = 4.76$, $p = .01$, $\eta_P^2 = .04$. Once again, the interaction's effect size is small and post hoc tests did not reveal any differences between learning groups because the interaction resulted from small differences between switch types across time (see S1 Data).

Since learning distance did not play any significant effect on the percentage of taxonomic generalization choices and that its effect on reaction times was small, and because of its absence of role in the eye tracking measures we removed learning distance from further analysis to simplify the design and improve the robustness of our results. We kept generalization distance in the analyses.

**Fixation times.** First, we analyzed the proportion of fixation times allocated to the AOIs during trials. Following standard practice [74, 75], a gaze towards an AOI was considered as a fixation when both eye positions were recorded inside the AOI's boundaries for longer than 60ms. Because our sample size was large, we checked that the data was normally distributed with Q-Q plots by group that confirmed that proportions of fixation times met the normal distribution requirements for ANOVAs.

We ran a four-way ANOVA on the proportion of fixation times in correct trials, with age (5, 6, 8 years) as a between-factor and generalization distance (near, distant), time slice (beginning, middle, end) and AOI (L, Th, Ta, P) as within-factors. The analysis revealed a simple effect of AOI, $F(3, 366) = 74.50$, $p < .001$, $\eta_P^2 = .38$. The analysis also revealed interaction effects between generalization distance and time slice, $F(2, 244) = 3.27$, $p = .04$, $\eta_P^2 = .03$, and between time slice and AOI, $F(6, 732) = 85.54$, $p < .001$, $\eta_P^2 = .41$ (see Fig 6). A three-way interaction was also revealed between age, time slice and AOI, $F(12, 732) = 2.40$, $p < .01$, $\eta_P^2 = .04$.

The triple interaction had a small effect size ($\eta_P^2 = .04$). A post hoc Tukey test revealed no significant difference between age groups. All the observed differences were between AOI and time slice and were redundant with those revealed by the two-way interaction. To simplify, we

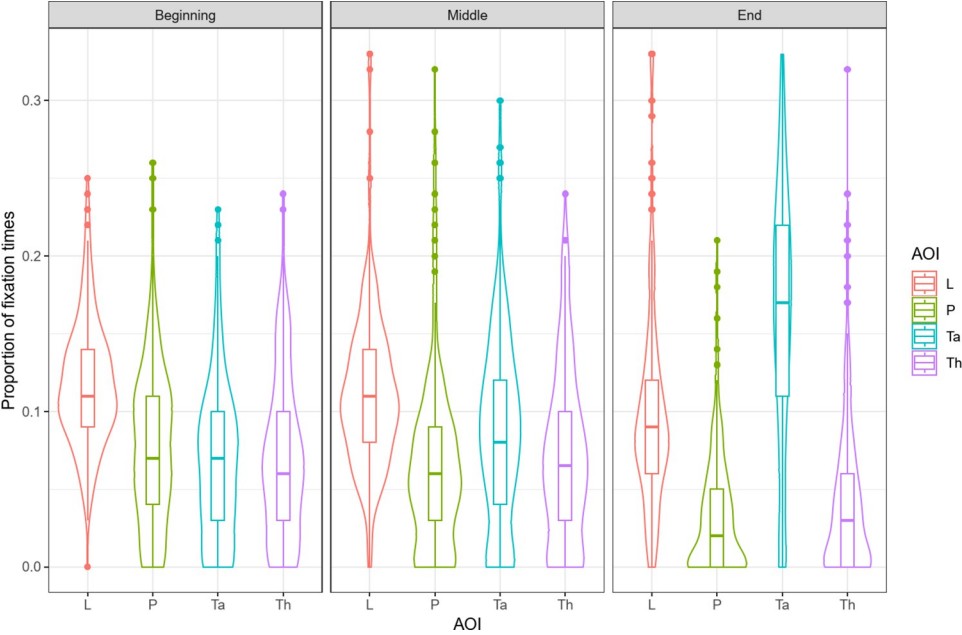

**Fig 6. Proportion of fixation times for correct answers as a function of time slice (beginning, middle, end), and Area of Interest (L, Th, Ta, P).**

therefore focused on the two-way interaction between time slice and AOI (see Fig 6). We ran contrast analyses using Bonferroni corrections to analyze this interaction. The contrasts of interest were those between different AOIs during the same time slice and those between same AOI across different time slices. The Bonferroni-adjusted α-value threshold was set at $p = .0016$.

The interaction reveals that at the beginning of a trial, children looked significantly more at the learning items than at the other items (L > Th $p < .001$, L > Ta $p < .001$, L > P $p < .001$). In the middle slice, they still looked more at the learning items (L > Th $p < .001$, L > Ta $p < .001$, L > P $p < .001$). At the end of a trial, they looked more at the taxonomic option (Ta > L $p < .001$, Ta > Th $p < .001$, Ta > P $p < .001$) and also looked more at the learning items than the distractors (End: L > Th $p < .001$, L > P $p < .001$). Taken together, these contrasts show that children started by looking at the learning items and continued to do so throughout the whole trial. While they also looked at all the available generalization options at the beginning of the trials, they gradually turned away from the distractors (i.e., Th and P) in order to concentrate their attention on the taxonomic option.

Contrasts for each AOI across slices reveal that children's looking times to the learning items decreased significantly between the middle and the end of the trials (L: Middle>End $p < .001$). The time spent looking at the thematically related distractor decreased between the middle and the end of each trial; it increased for the taxonomic item and decreased for the perceptual distractor as the trial progressed (Th: Middle > End $p < .001$; Ta: Beginning < Middle $p < .001$, Middle < End $p < .001$, P: Beginning < End $p < .001$). These contrasts confirm that, as each trial progressed, children's attention moved away from the distractors and towards the taxonomic item.

**Switches.**　Following common practice [74, 75], switches were defined as movements from one AOI to another, provided that children's gaze stayed focused on each AOI for at least 60 ms. Since there were 5 stimuli, which could a priori result in 20 different switches, we aggregated them in five different switch types that were the most relevant for our analysis. Thus, LL switches are all switches between $L_1$ and $L_2$, LTh switches are switches between $L_1$ or $L_2$ and Th and vice versa. Similarly, LTa switches are switches between $L_1$ or $L_2$ and Ta and vice versa; LP switches are those between $L_1$ or $L_2$ and P and vice versa; and ThTaP switches are all switches between the available generalization options (Th, Ta and P).

We analyzed the log transformation of the number of switches to fit the normality assumptions for ANOVAs. We ran a four-way ANOVA on data from correct trials, with age (5, 6, 8 years) as a between-factor and generalization distance (near, distant), time slice (beginning, middle, end) and switch type (LL, LTh, LTa, LP, ThTaP) as within-factors. The analysis revealed simple effects of age, $F(2,108) = 8.82$, $p < .001$, $\eta_P^2 = .14$, generalization distance, $F(1,108) = 24.03$, $p < .001$, $\eta_P^2 = .18$, time slice, $F(2,213) = 73.09$, $p < .001$, $\eta_P^2 = .40$, switch type, $F(4,432) = 40.92$, $p < .001$, $\eta_P^2 = .28$. The effects of age and generalization distance showed that 5-year-old children made less switches than 6- and 8-year-old children ($M_5 = 0.34$; $M_6 = 0.46$; $M_8 = 0.47$; $5 < 6$, $p < .001$) and that children made less switches in the distant generalization setting than in the near generalization setting ($M_{Near} = 0.46$; $M_{Distant} = 0.38$; $p < .01$).

The most interesting result involving switch type and time slice was the interaction effect between time slice and switch type, $F(8,864) = 11.24$, $p < .001$, $\eta_P^2 = .09$, as it reveals the temporal dynamics of children's searches across switches (see Fig 7).

We ran contrast analyses with a Bonferroni correction set at $p < .0011$. The contrasts of interest were those between different switch types during one and the same time slice, and those between the same switch types over different time slices. The interaction reveals that, at the beginning of a trial, there were significantly more LP switches than LL switches ($p < .001$) and more LTh switches than LL switches ($p < .001$), showing that children's attention was initially attracted to the distractors.

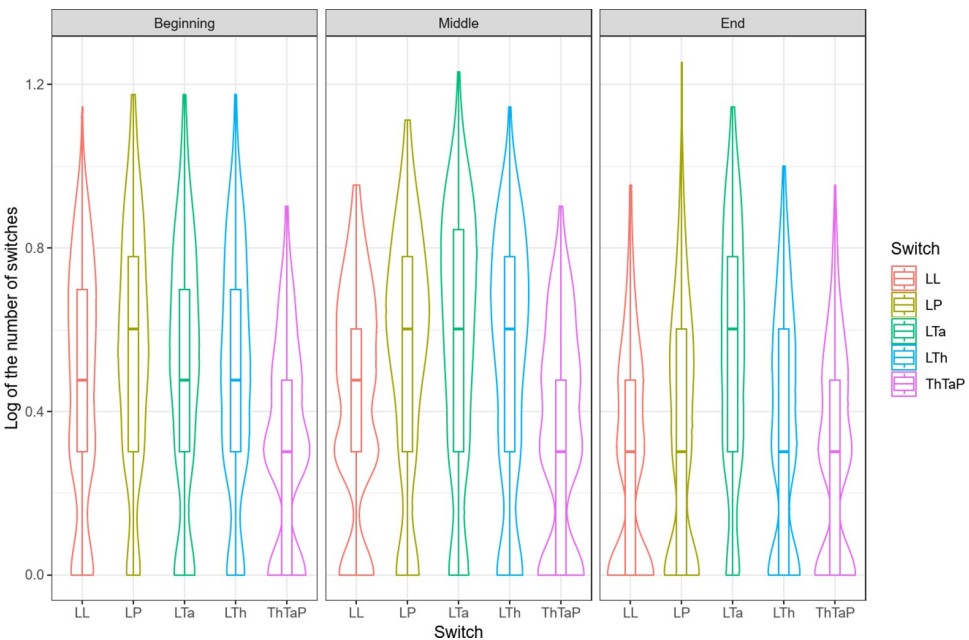

**Fig 7. Log of the number of switches for correct trials as a function of time slice (beginning, middle, end), and switch type (LL, LTh, Lta, LP, ThTaP).**

At the beginning of each trial, children also made less ThTaP switches than all the other switch types ($p < .001$). In the middle, there were more switches between learning items and available generalization options (i.e., LTh, LTa, LP) than between learning items themselves (i.e., LL) (LTh > LL $p < .008$, LTa > LL $p < .001$, LP > LL $p < .001$). There were also fewer switches between available generalization options (i.e., ThTaP) than any other type of switch in the middle of the trials (ThTaP < LL-LTa-LTh-LP, $p < .001$). At the end of the trials, children made significantly more LTa switches than all the other switch types, which did not differ from one another (LTa > LL $p < .001$; LTa > LTh $p < .001$; LTa > LP $p < .001$; LTa > ThTaP $p < .001$).

These contrasts clearly show that children started by comparing all the items with one another, showing a slight preference for LP switches at the beginning of the trials, likely due to the salience of the perceptual distractors. While they then continued to make comparisons between learning items (i.e., LL switches) throughout each trial, they made less switches towards the distractors (i.e., LTh and LP) in the middle and at the end of the trials. This shows that they turned away from these distractors and gradually made more LTa switches.

Individual contrasts run for each switch type across time slices reveal that the number of LL switches decreased at the end of each trial (Middle > End, $p < .001$). Not surprisingly, the same switching time-course was found for LTh switches (Middle > End, $p < .001$) and for LP switches (Middle > End, $p < .001$). These contrasts show that children systematically explored the three options in the middle of each trial but turned away from the distractors during the trial as they reached a taxonomic decision.

**Correct (taxonomic) answers versus errors (distractors).** For these analyses we analyzed 6-year-old children's data only, since 5-year-old children had low scores, with too few correct answers, The reverse was true for 8-year-old children. We analyzed the log of the number of switches to fit ANOVA normality requirements.

We ran a four-way ANOVA on the proportion of fixation times from all trials from six year old children, with trial accuracy (correct, error), generalization distance (near, distant), time

slice (beginning, middle, end) and AOI (L, Th, Ta, P) as within factors. We applied Green-house-Geisser or Huynh-Feldt corrections when necessary. Here, to identify the differences between gaze profiles that led to correct answers and those that led to errors, we focus on inter-actions with trial accuracy (full results of this analysis can be found in S2 Data). The analysis revealed three interactions including trial accuracy. These were the interactions between trial accuracy and AOI $F(3,105) = 8.35$, $p < .001$, $\eta_P^2 = .19$, between trial accuracy, generalization distance and AOI $F(3,105) = 3.03$, $p = .03$, $\eta_P^2 = .08$ and finally between trial accuracy, time slice and AOI $F(6,210) = 16.21$, $p < .001$, $\eta_P^2 = .32$. We ran post hoc Tukey analyses, and, not sur-prisingly differences were revealed between proportion of fixation times to the taxonomic item in the end of trials in both the interaction between trial accuracy and AOI ($p_{Tukey} < .05$,) and the interaction between trial accuracy, time slice and AOI ($p_{Tukey} < .001$). Proportion of fixa-tion times to the taxonomic item were higher in correct trials than in error trials (i.e., choice of one of the distractors): $M_{Correct} = 0.09$ and $0.13$, $M_{Error} = 0.06$ and $0.04$.

Similarly, we ran a four-way ANOVA on the log of the number of switches from all trials with trial accuracy (correct, error), generalization distance (near, distant), time slice (begin-ning, middle, end) and switch type (LL, LTh, LTa, LP, ThTaP) as within factors. We applied Greenhouse-Geisser or Huynh-Feldt corrections when necessary. Here, to identify the differ-ences between gaze profiles that led to correct answers and those that led to errors, we focus on interactions with trial accuracy (full results of this analysis can be found in S2 Data). The analysis revealed an interaction between trial accuracy and time slice $F(2,52) = 6.02$, $p < .01$, $\eta_P^2 = .18$, and an interaction between trial accuracy and generalization distance $F(1,28) = 7.71$, $p = .01$, $\eta_P^2 = .22$, and an interaction between trial accuracy, time slice and switch type $F(8,224) = 2.53$, $p = .01$, $\eta_P^2 = .08$. Post hoc analyses (contrasts) did not reveal any significant differences between correct and error profiles. This suggests that errors and correct answers did not differ in the general search profile. We now examine the possibility that it might be because the defi-nition of the time slice was too coarse.

**First gazes: Distribution and predictive power.** The analyses above reveal that all AOI were gazed at and that all switch types were produced in the first time slice. Hence, the pattern observed during the first slice might have obscured regularities in gaze sequences between learning items and between learning items and options in the early gazes of the trial. The absence of clear difference between correct (taxonomic) and errors (distractors), specifically in the first time slice might have also resulted from these early gazes towards all the stimuli. For this reason, we analyzed the first five fixated items. Given our research hypotheses, we consid-ered whether children would first gaze at the learning items, aligning them in order to extract their common properties, or whether they would first look at the generalization items, especially towards the shape similar distractor. We hypothesized that more gazes towards the learning items rather than the options would be predictive of correct trials. To test this hypothesis, we calculated a ratio measuring the orientation of participants' attention for each of the first five gazes. This ratio was defined as the number of gazes towards the learning items made by a par-ticipant divided by the number of gazes directed towards the generalization items by the same participant and was named Learning-items/Generalization-items (hereafter Learn/Gen). Higher values of the ratio mean that children looked more at the learning items, and that a participant focused more on the learning items for this particular gaze rank (1st, 2nd, etc.). This kind of inte-grative measure is very useful as it aggregates the strategy adopted by participants under a single value and powerful hypotheses can be derived from it. First, if our hypothesis that participants will first fix their gaze on learning items is highly predictive of performance [67, 76], we predict that older children will have higher ratio values for the initial gazes (e.g., 1st and 2nd) and that higher values for these gazes will be correlated with a higher number of taxonomic choices.

**Table 1. The goodness of fit of the generalized linear models with children's generalization scores (in near generalization cases) as the outcome and Learn/Gen ratios as predictors.**

| Outcomes | Model | Predictors | Df | AIC | $R^2$ | Adjusted $R^2$ | p-value |
|---|---|---|---|---|---|---|---|
| Percentage of correct answers | M0 | Age + Learning | 1 | 1338 | .163 | .151 | |
| | M1 | Age + Learning + Learn/Gen$_1$ | 1 | 1302 | .362 | .348 | < .001 |
| | M2 | Age + Learning + Learn/Gen$_1$ + Learn/Gen$_2$ | 1 | 1295 | .401 | .383 | < .01 |
| | M3 | . . . + Learn/Gen$_3$ | 1 | 1292 | .420 | .399 | < .05 |
| | M4 | . . . + Learn/Gen$_4$ | 1 | 1293 | .425 | .418 | .29 |

The model controls for age and learning distance.

We ran a three-way repeated-measures ANOVA on the Learn/Gen ratio with age (5, 6, 8-years old) as a between-factor and generalization distance (near, distant) and gaze position (1, 2, 3, 4, 5) as within-factors. No corrections to the data were needed. The analysis revealed effects of age, $F(2, 121) = 3.20$, $p = .044$, $\eta_P^2 = .05$ ($M_5 = 1.01$, $M_6 = 1.31$, $M_8 = 1.26$, 5yo < 6yo $p = .036$), generalization distance, $F(1,121) = 8.02$, $p < .01$, $\eta_P^2 = .06$ ($M_{Near} = 1.29$; $M_{Distant} = 1.10$), and gaze position, $F(4, 484) = 17.56$, $p < .001$, $\eta_P^2 = .13$ ($M_1 = 1.76$; $M_2 = 0.94$; $M_3 = 1.18$; $M_4 = 0.96$; $M_5 = 1.14$). Post hoc Tukey tests reveal that the Learn/Gen ratio was significantly higher for the first gaze than for each of the other gazes ($p < .001$). For the following ratios (gaze 2 to 5), there was no significant difference between gazes. No interaction reached significance ($p > .05$). Thus, the ratio is seen to increase with age, a finding which is compatible with our hypothesis that the ability to make comparisons increases with age. We also found that the ratio decreased after the first gaze, meaning that children first looked at the learning items but then was distributed equally between the learning items and the options.

We next tested whether early attention orientation was predictive of children's taxonomic generalization performances [39, 67]. To this end, we first computed Spearman correlations, for correct trials, between participants' Learn/Gen ratios and their percentage of taxonomic answers. The correlations revealed a positive correlation between the ratio value and the number of correct trials for the first five gazes (see S3 Table).

We then used general linear models with children's generalization scores (i.e., proportion of taxonomic generalizations) as outcome and Learn/Gen ratios as predictors, while controlling for age and learning distance. Our best-fit models (M3 for near generalization, M4 for distant generalization) included the Learn/Gen ratio for the first three gazes in near generalization cases (see Table 1), and the Learn/Gen ratio for the first four gazes in distant generalization cases (see Table 2). We ran ANOVAs on the best fitting models (see S4 and S5 Tables).

**Table 2. The goodness of fit of the generalized linear models with children's generalization scores (in distant generalization cases) as the outcome and Learn/Gen ratios as predictors.**

| Outcomes | Model | Predictors | Df | AIC | $R^2$ | Adjusted $R^2$ | p-value |
|---|---|---|---|---|---|---|---|
| Percentage of correct answers | M0 | Age + Learning | 1 | 1207 | .058 | .043 | |
| | M1 | Age + Learning + Learn/Gen$_1$ | 1 | 1158 | .370 | .355 | < .001 |
| | M2 | Age + Learning + Learn/Gen$_1$ + Learn/Gen$_2$ | 1 | 1152 | .407 | .388 | < .01 |
| | M3 | . . . + Learn/Gen$_3$ | 1 | 1143 | .456 | .433 | < .001 |
| | M4 | . . . + Learn/Gen$_4$ | 1 | 1139 | .484 | .458 | < .05 |
| | M5 | . . . + Learn/Gen$_5$ | 1 | 1138 | .495 | .466 | .102 |

The model controls for age and learning distance.

These results strengthen our hypothesis that an initial focus on the learning items, and therefore not studying the options before interpreting the learning items, predicts taxonomic generalization in both near and distant generalization conditions. It is interesting to note that successful generalization in the distant generalization condition was predicted by a longer string of gazes towards learning items (four gazes) than in the near generalization condition (three gazes). This is consistent with the idea that children needed to compare the learning items more frequently in the more difficult condition in order to successfully align the learning items, with the result that longer strings remained predictive.

## Different strategies for different performance? High achievers versus low achievers

The previous analysis revealed that initial gazes were highly predictive of children's scores. In particular higher Learn/Gen ratio values predicted a higher percentage of taxonomic responses. However, the fact that this ratio for earlier gazes significantly predicted the percentage of correct generalizations does not mean that high ratios necessarily equate to high performance levels. Thus, a posteriori from previous analysis we suggested to explore the relation between children's early gaze orientation and their levels of achievement. The present analysis complements the previous ones in this exploratory attempt. It also provides another look on the difference between correct trials (taxonomic) and errors (distractors), which did not reveal important differences between them. Our general hypothesis considered that higher-performing children might follow different strategies depending on the chosen option, taxonomic or perceptual/thematic. This means that the choice they eventually make depends on what they do throughout the entire trial, before coming to their decision. However, results from previous studies suggest that it is also possible that the early search steps prepare for both correct and incorrect choices. It has also been shown that early gazes and/or switches have reliably predict the outputs (error or correct response) in analogy tasks [67, 76]. On the other hand, errors might result from the fact that children finally opted for one of the two non-taxonomically related choices at the decision stage despite having followed the same search strategy in both correct and incorrect trials. Hence, we tested whether high achievers and low achievers followed the same sequence of gazes at the beginning of the trial in order to explore early gazes' relation with generalization performances and confirm (or disconfirm) the importance of early comparisons for correct generalization. Indeed, the last steps in the decision-making process are less interesting since, by definition, low achievers choose more distractors than high achievers and therefore gaze more at them and less at the taxonomic match.

High-achieving children were defined as those who obtained eleven or more correct answers (over 14 trials) and low-achieving children as those obtaining four or less correct answers. This measure ignores participants characterized by ambiguous profiles, with less contrasted strategies (if any). We analyzed 6-year-old children's data only, since 5-year-old children had low scores with too few "high-achieving" children, whereas the reverse was true for 8-year-old children. Among the 6-year-old children, there were 30 high-achieving children and 18 low-achieving children.

We analyzed the Learn/Gen ratio described above. We hypothesized that high-achieving children would focus more on the learning items at the beginning of the trial, with a higher Learn/Gen ratio than low-achieving children. We ran a 3-way repeated-measures ANOVA on the Learn/Gen ratio computed on the first five gazes for correct answers, with subject profile (low, high) as a between-factor, and generalization distance (near, distant) and gaze position (1, 2, 3, 4, 5) as within-factors. Results revealed significant effects of participant profile, $F(1,42)$ = 35.4, $p < .001$, $\eta_P^2 = .46$, with low-achieving subjects' ratios being significantly lower than

those of high-achieving subjects, $M_{Low}$ = 0.80, $M_{High}$ = 1.61. They also revealed a main effect of gaze position, with the Learn/Gen ratio being significantly higher for the first than for the second and fourth gaze positions, $M_1$ = 1.79, $M_2$ = 0.92, $M_3$ = 1.07, $M_4$ = 0.92, $M_5$ = 1.32, (t-test with Bonferroni-corrected significance level set at $p$ = .005, $p$ < .005). The reason why M2 and M4 were lower than M1 but not M3 and M5 is not entirely clear. Starting with a learning item, some participants gazed at another learning item, whereas others tended to gaze back and forth between learning and transfer items. No other effect reached significance ($p$ > .05).

These two effects reveal that high-achieving children focused significantly more than low-achieving children on learning items than on the possible options during their initial gazes, confirming previous results [67]. This result also makes sense in terms of the strategies: children who followed our third strategy, the goal-directed strategy [67], would have subsequently been at risk of choosing the perceptual distractor, rather than the option resulting from a careful alignment of the learning items, namely the taxonomic option.

## Discussion

Building on earlier results on novel noun learning obtained by Gentner and Namy, who used a comparison design, we proposed that this ability can be more generally described in terms of the alignment of the learning stimuli. Our main goal was to use an approach similar to that adopted in other analogical reasoning studies [67, 76, 77] in order to provide a description of the temporal dynamics of children's learning/generalization strategies when making comparisons. The main aims were to (1) provide a description of novel noun generalization strategies in a learning-with-comparison design in terms of patterns of gazes and switches during the trial; (2) interpret this data in terms of a set of hypothetical strategies; (3) assess the organization of the search and how it interacts with age and task difficulty (i.e., conceptual distance between items); (4) assess whether early gazes predict generalization performance; (5) provide a description of strategies that lead to high performance (good achievers) or poor performance (low achievers). Overall, our results reveal that the three age groups had gaze and switch profiles which were compatible with an Elimination strategy. Early attention to distractors (e.g., shape) seems to be correlated with poor performance, whereas early alignments of learning items lead to better performance. Differences observed between strategies are discussed in an executive function framework.

### Connecting eye movements with strategies

The focus is placed on the time course of gazes and switches during a trial. The important results are, first, that the fixation times for the learning items were longer than those for the generalization options at the beginning of the trials. Switches between the learning items were also very frequent at the beginning of each trial. This suggests that the learning items play an early anchoring role in novel noun generalization strategies. The importance of these early gazes towards the learning items is also shown by their predictive power, as indicated by the analyses of the initial gazes.

Second, gazes towards the generalization options and switches between learning items and generalization options appeared later in the trials, while there were far less switches between the available generalization options themselves. Thus, switches between learning and generalization options constituted the main pattern exhibited by children when focusing on a generalization solution.

The study's second aim was to test this data set against four possible generalization strategies, the "Elimination strategy", the "Construction strategy", the "Goal-Directed" strategy and the "Salient Shape strategy". Both the Elimination and the Construction strategies are

compatible with an early focus (longer gazes and switches) on the learning items, which we interpret in terms of a search for and comparison of common features. The Goal-Directed strategy is less compatible with this early focus on the learning items because it predicts an early focus on the generalization items. An Elimination strategy calls for a large number of switches between learning items and available generalization options, which is what was observed. A Construction strategy would have predicted switches between available generalization options, that is comparisons of the generalization options with each other, which was not the case or, at least, much less so than the case of switches towards options.

The Goal-Directed strategy predicted an equal distribution of gazes between learning items and generalization options, or an early focus on generalization options (longer gazes towards generalization options than other items and more switches between generalization options than other switches) which, again, was not the dominant pattern. The pattern of answers is also less compatible with the Salient Shape hypothesis, which describes a bottom-up process driven by shape similarities. Indeed, this view predicts early gazes and switches towards the perceptually similar items, whereas our results show that the perceptual distractor was the object of shorter gazes than the learning items.

However, in the beginning of trials, the results also revealed more switches between learning items and the perceptual distractor than between learning items themselves and this might cast doubt on our interpretation. As predicted by the alignment view, perceptual distractors might attract early comparisons because of their salience. However, successful trials were characterized not by longer *gazes* at the perceptual distractor–which would have been a sign of a bottom-up process—but by longer gazes towards learning items. However, the switches between learning items and the perceptual distractor co-occurred with switches between the learning items and the thematically-related distractor and the taxonomic option. These switches, which persisted until the taxonomic option was chosen over the other options, are compatible with a top-down decision process as the discussion of initial gazes will show.

Overall, the pattern of results is consistent with the response elimination strategy described for analogical reasoning [63, 64], in which items are aligned and eliminated until the best match is the only one left. It is also compatible with the "alignment-first" conception derived from the alignment hypothesis [53], which predicts that each option will be aligned until one is kept as the solution, either correct or incorrect.

## Profiles and predictive power of the early gazes

Our fine-grained analysis of the children's first five gazes revealed two important and related results. The first is that the ratio between learning-stimulus gazes and generalization-stimulus gazes (Learn/Gen) was predictive of taxonomic generalization performance up to the first three gazes in near generalization trials and up to the first four gazes in distant generalization trials. Hence, a high ratio of early gazes towards the learning items is predictive of a taxonomic choice. Again, this makes sense in terms of alignment and confirms the previous analyses run on gazes and switches. These early alignments between learning items are important in helping to build a deeper encoding of the learning items which, in turn, makes it possible to find the taxonomic option. This finding echoes those reported for analogical tasks that have also revealed the importance of early gazes towards the base domain items (A and B stimuli) before participants start to search for the task solution among the available generalization options [77, 78]. In both cases, it is interesting that children's ability to control their search and fixate learning items not only improves with age, as was the case here, but is also positively correlated with success.

The second important and related result is that high-achieving children were able to selectively concentrate more on the learning items at trial onset than low-achieving children. This

was confirmed by the higher Learn/Gen ratios for high-achieving children across the five gazes and supports the interpretation that achieving correct generalization may be dependent on children's ability to efficiently control their strategy by focusing on the learning items.

This result is analogous to results obtained in analogical reasoning tasks. As in the present experiment, the eye-movement profiles of the low-achieving participants in Thibaut and colleagues' studies on analogies (see also [79, 80]) were more biased towards the C item and the available options (C and the options defining the goal of the task) than those of high-achieving participants. Indeed, as shown by [81], when the analogy task required children to first analyze and identify the relation between A and B before they were allowed to see C and the options (C and D part of the analogy), the results were better than in a condition in which all the stimuli composing the analogy were shown simultaneously [82]. Thus, anchoring generalization in a thorough analysis of the learning items seems to be the first requirement before then comparing the learning items with the generalization items. In terms of strategy, this suggests that participants who follow a bottom-up strategy which requires them to attend to the shape (or other distractors) are less efficient. This is exactly what the analyses on the first gazes show. The same is true for those who follow the Task Goal-Directed strategy and distribute their attention across all the stimuli, including the generalization options, early on, at the risk of diluting task encoding by including all the stimuli rather than just the learning items. The latter hypothesis is interesting as it provides an interpretation in terms of processes (inhibition of the task's main goal) rather than in terms of content, that is in terms of what information is used (perceptual or deeper, or both). These two strategies are compatible with the profiles of children who directed more early gazes towards distractors (low achievers) and who might be those who failed to inhibit this information and were less successful in the task. In this respect, an analysis of six-year-old children's errors (paired t-test) revealed that they made significantly more perceptual errors than thematically related errors ($M_P$ = 38.58%, $M_{Th}$ = 11.35%, $t(184)$ = 4.10, $p < .001$, d = 0.47). This argues in favor of the interpretation that the children who made mistakes were the ones who chose the perceptual distractor instead of integrating all the provided information. These participants are those who followed the Salient-shape strategy.

## Are generalization strategies mediated by age and task difficulty?

We manipulated age and conceptual distance as two factors known to influence generalization performance and which are thus potential mediators of generalization strategies. Behavioral data confirmed that both age and distance could play a role. Eye-tracking data revealed a main effect of age and of generalization distance, even though these two factors were not involved in any interaction with other factors. For example, data did not reveal a switch from elimination to construction strategies as task difficulty increased.

Age and generalization distance do, however, affect generalization performance. In the distant generalization condition (i.e., when the difficulty of the task increased), younger children made significantly more errors or had higher generalization RTs. In terms of strategies, there were main effects of age and generalization distance on the number of switches, with less switches for younger children and for distant generalization items. This might appear paradoxical since it might be expected that younger children would need to make more comparisons in order to find a solution and that distant items would require more comparisons. The latter result might suggest that younger children were unable to control the task until they had achieved a relevant encoding of the learning items and, indeed, they probably gave up attempting to do this earlier. This is consistent with their near-chance performance and the fact that they answered more rapidly in the difficult cases. Similarly, the more difficult, distant, generalization condition may also have elicited less comparisons overall (log of the number of

switches: $M_{Near} = 0.46$, $M_{Distant} = 0.38$, $p < .001$, see results for more details), most likely because children found it difficult to unify the items. Finally, the models of the predictive power of the Learn/Gen ratio for the initial gazes in the two generalization distance conditions showed that a longer series of gazes was more predictive in the distant generalization condition than in the near generalization condition (first four gazes were predictive versus first three gazes in the case of near generalization). Thus, it seems that it is not the number of gazes per se that matters for selecting the taxonomic match but rather the fact that they are targeted on learning items.

In sum, at any age, being a smart generalizer starts with systematic alignments of the learning items at the beginning of the trial, followed by systematic comparisons with the options later on. Comparing the options with one another, as would be predicted by a projection-first strategy or construction strategy (see above), does not seem to be the most important factor here, even though children did make these comparisons. Previous contributions on comparison designs have described novel noun learning as an alignment between learning items [37, 47]. This view is important but neglects the steps that occur after or in parallel with alignment, namely the steps involving comparisons with the options. Generalization also involves successive comparisons, as each option provides a set of possibilities for new alignment hypotheses. During generalization, children apply the vocabulary of dimensions that they have abstracted during learning, keep it in mind and resist irrelevant options. In other words, they start with learning items and then generalize the dimensions to the new items. The case of the classical single novel noun learning task is very different in this respect. Children use their lexical biases to target certain dimensions and find the stimuli that are similar along this dimension. It is more a "check and decide" process than the constructive process described here.

## Limits of the present study

We focused on a small number of stimuli that we thought were representative of the issue. This has been the strategy followed by most developmental studies and seems a sensible one to replicate. In a previous study [67] small subsets of the data were found to leed to the same strategies (near perfect correlation) as the full data set. Second, our experiment focused on object nouns. The extent to which our results can be generalized to other types of nouns, such as relational nouns [43], remains a matter of debate. Indeed, relational nouns involve at least two entities (e.g., being the "neighbor" of, "addition" of numbers). Thus, participants would have to analyze the two related entities and align the output of this comparison with the second set of stimuli that implement the same relational noun. Third, it would be interesting to study younger children. However, with the materials and the eye-tracker used here, we could not collect reliable data from younger children. Despite this, extending the age range remains a desirable objective.

## Conclusion

The main purpose of the present study was to describe children's generalization strategies and assess how these may be mediated by task difficulty. Overall, our results give a fairly clear picture of what a successful alignment looks like for children aged from five to eight years. Children mostly use what can be described as an elimination strategy, in which they first look longer at the learning items and make comparisons between them. Also, switches between learning items and available generalization options start very early in the task, as some are observed at the same time as the comparisons between learning items at the beginning of the trials. This dominant strategy was used in a robust way, whatever the age and task difficulty. Differences in generalization can therefore not be explained by strategy profile differences

here. Two important aspects have been shown to determine the success of conceptual generalization: the search pattern children use to navigate towards the solution and their early attention to learning items rather than available generalization options.

## Supporting information

**S1 Table. Full list of materials.**
(DOCX)

**S2 Table. Similarity ratings by category.**
(DOCX)

**S3 Table. Results of spearman correlations.**
(DOCX)

**S4 Table. Details of the ANOVA run on M3 model.**
(DOCX)

**S5 Table. Details of the ANOVA run on M5 model.**
(DOCX)

**S1 Data. Supplementary results 1.**
(DOCX)

**S2 Data. Supplementary results 2.**
(DOCX)

**S3 Data. Full data set.**
(XLSX)

## Acknowledgments

The first author benefited from a PhD scholarship from the Ministry of Higher Education and Research. The authors also thank Yannick Lagarrigue and members of the LEAD laboratory for helpful discussions.

## Author Contributions

**Conceptualization:** Arnaud Witt, Jean-Pierre Thibaut.

**Data curation:** Eleanor Stansbury, Patrick Bard.

**Formal analysis:** Eleanor Stansbury.

**Funding acquisition:** Eleanor Stansbury, Arnaud Witt, Jean-Pierre Thibaut.

**Investigation:** Eleanor Stansbury.

**Methodology:** Eleanor Stansbury, Arnaud Witt, Jean-Pierre Thibaut.

**Project administration:** Eleanor Stansbury.

**Resources:** Eleanor Stansbury, Arnaud Witt, Jean-Pierre Thibaut.

**Software:** Patrick Bard.

**Supervision:** Arnaud Witt, Jean-Pierre Thibaut.

**Validation:** Arnaud Witt, Jean-Pierre Thibaut.

**Visualization:** Eleanor Stansbury.

**Writing – original draft:** Eleanor Stansbury, Arnaud Witt, Jean-Pierre Thibaut.

**Writing – review & editing:** Eleanor Stansbury, Arnaud Witt, Jean-Pierre Thibaut.

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
