## [Decision Letter · Decision Letter 0]

29 Mar 2023

PONE-D-22-35572How children generalize novel nouns : an eye tracking analysis of their novel noun generalization strategies.PLOS ONE

Dear Dr. Stansbury,

Thank you for submitting your manuscript to PLOS ONE. After careful consideration, we feel that it has merit but does not fully meet PLOS ONE’s publication criteria as it currently stands. Therefore, we invite you to submit a revised version of the manuscript that addresses the points raised during the review process. Both reviewers agreed that your manuscript would benefit from being positionned in a broader context.I also want to add that bar graphs with error bars (Figures 2-3-4-5-6) should be avoided as they hide the underlying distribution of the data. Please replace them with richer representation of the data such as box plots overlayed on jittered individual data points, violin plots, or raincloud plots. There are many open access and ready-to-use toolboxes, e.g. https://wellcomeopenresearch.org/articles/4-63

We look forward to receiving your revised manuscript.

Kind regards,

Antoine Coutrot

Academic Editor

PLOS ONE

Journal Requirements:

5. We note that you have referenced [ie. Thibaut, J. P. (1991)] which has currently not yet been accepted for publication. Please remove this from your References and amend this to state in the body of your manuscript: (ie “Thibaut, J. P.[Unpublished]”) as detailed online in our guide for authors:

6. Please include a separate caption for each figure in your manuscript.

Reviewers' comments:

Reviewer's Responses to Questions

**Comments to the Author**

1. Is the manuscript technically sound, and do the data support the conclusions?

Reviewer #1: Partly

Reviewer #2: Yes

2. Has the statistical analysis been performed appropriately and rigorously? 

Reviewer #1: I Don't Know

Reviewer #2: Yes

3. Have the authors made all data underlying the findings in their manuscript fully available?

Reviewer #1: No

Reviewer #2: Yes

4. Is the manuscript presented in an intelligible fashion and written in standard English?

Reviewer #1: Yes

Reviewer #2: No

5. Review Comments to the Author

Reviewer #1: The question of how children explore and categorize is an important one with implications for multiple research areas. The current approach is also innovative and has potential to inform our understanding of children’s generalization abilities, especially because it integrates multiple variables. I was excited to read the study because of its potential. However, that excitement was dampened for a number of reasons. First, the current manuscript is written for a specific niche group of researchers – those that align with the alignment/comparison framework and come from the Gentner/Thibaut school of thought. Because of this, the impact of the current study is diminished and the manuscript would benefit from broader framing and wider inclusion of both theoretical and empirical literature. At the same time, better operationalization of some of the concepts (e.g. how is “aligning” operationalized, how is “semantic distance” measured) is also needed as are missing details in the methods and rationale for the analytic decisions. Overall, the data and question are interesting and relevant, but the particular framing, methods used, and conclusion drawn do not make that relevance clear or broadly applicable. Specific comments for each section are detailed below.

Introduction:

The introduction is a bit disjointed and some of the theoretical background is long-winded without a clear connection to the current study. For instance, the section on analogical reasoning (beginning end of page 5) assumes the reader is intimately familiar with how analogical reasoning and its methods are different from other generalization/categorization approaches described before. When the current study is discussed on pages 7-8 in terms of analogical strategies, exactly how “alignment” is being defined or measured or how the researchers will know when a child has “found a common property” is unclear, which makes the relevance of these strategies even less useful. Relatedly, what if the process of categorization is not based on the conceptual ideals analogical reasoning relies on? For instance, on page 5 (line 94), it is noted that “nouns contribute to the alignment process, most likely by inviting one to engage in the comparison process.” (page 5, line 94). While the evidence does suggest that nouns bias children’s attention, there is still much debate about if it actively “invites alignment” or if the process is conceptually-driven or lower-level attention-driven (c.f. Smith & Samuelson, 2006). More inclusion of work highlighting perceptual properties (e.g. shape bias) or empirical work using looking based measures for generalization-relevant tasks (e.g. Yoshida et al., 2019) would help.

Methods:

The methods are particularly brief and limited on details as well as rationale. Can the authors comment on what extend using 2D vs. 3D stimuli matters as well as how using familiar referents (vs. entirely novel items such as those used in classic shape bias work) might impact the results? It would seem that the former (2D images) limits the use of perceptual features as one does not have access to things such as material or texture. Likewise, the use of known referents would bias individuals to use taxonomic or thematic relations over perceptual ones, given extensive history with those items. In regards to the use of familiar items then, capturing children’s prior experience with those items and knowledge of their real labels would be relevant. The authors note on page 15 that “participants were supposed to know the items” and “percentage of unknown items was very low” but no statistics or details for this specific sample are given.

More details also need to be provided in the methods in order to replicate the study. For instances, what is the timing for presentation of the stimuli on each trial? Were all 5 pictures shown at once? At one point did the verbal directive start/end? How did the task direct children’s attention – e.g. the prompt is “This is a buxi”, but how did the child know what “this” referred to? Was it highlighted, pointed to by the experimenter, no directive (so referent might be unclear), etc.? How was the eye-tracking data cleaned and processed? Finally, how was a child’s “choice” determined and from what points were reaction time calculated?

If my interpretation of the results is correct, children switched from L to P more often at the start of the trial – this is entirely consistent with attention-based learning accounts (not just “salience”, line 492). Then, in the middle of the trials, children look to all test items, but by the end of the trials, they are looking more to Ta. However, knowing at what point the prompt/target word offset in the methods is critical – does the “early” time bin capture pre-word looking or post or both?

Results:

The analysis approach, on the surface, seems sound, but there is ambiguity as to why specific decisions were made, such as which variables to include or which sub-sample to run the analysis over. The lack of rationale for these decisions make interpreting the results difficult. As one example, both the reaction time analysis and the eye-tracking analysis were only run on trials that led to a correct choice. Was accuracy related to reaction time or looking? Why not run this analysis over all trials? There are also some possible inconsistencies (or at the very least, confusion) over results. For instance, on line 387, the authors note that learning distance did not have an effect on generalization, but on line 362, they note that near scores were higher than distance scores, suggesting that distance DID matter. By line 389, the authors note that learning distance and eye-tracking were not related, but no statistical evidence for this is given (and if reaction time was based on looking behavior, lines 381 would contradict this). Thus, it is not clear why learning distance was eliminated from the eye-tracking analysis (line 389), especially when it was one of the key apriori variables...but then again, it looks like learning distance was still included in the analysis (line 425)? In addition, further information is needed as to why the three-way interaction in looking behavior was dismissed (line 432). Finally, the analysis and discussion about low and high achievers seems unsupported, exploratory, and less connected with the rest of the study or analysis. This is particularly true because its not clear that the Learn/Gen ratio is the best way to capture this concept. In general, I would encourage the authors to consider mixed effects models for future work to better capture item and subject differences especially without having to create new composite ratios.

Discussion:

Similar to some of the comments on the introduction, there are some assumptions made about the interpretation of children’s behavior that may not be warranted. In particular, switching between the learning items does not necessitate that children are making a conceptually-driven alignment and comparison process. Rather, children may simply be habituating with the stimuli being highlighted. This is consistent with low-level memory and encoding processes and visual processing patterns. Similarly, the structure of the methods directs children to the learning items. Because of this, I would not expect a child to spend more time looking at the 3 test items and thus, support for a “construction” strategy is not likely to arise because of the methodological design. Because of this, I don’t think we can differentiate between the theories/strategies proposed and I don’t think the methods used are completely unbiased to do so. That being said, the data are still potentially interesting (pending details on the analysis as above) and it may be the case that we don’t need to posit one of these alignment strategies in order to inform our understanding of behavioral patterns in comparison tasks.

Minor points:

- First line of abstract reads a bit awkward and passive. I’m not sure that “comparison setting” is a term that would be familiar to all readers. Also, later in the abstract, the authors refer to learning items and test items, but it’s not clear what the difference is so the brief summary of the results is difficult to interpret.

- I was surprised to see that the 2002 4-step process view of the shape bias by Smith and colleagues was not cited, nor was work by Oakes and colleagues (e.g. Oakes & Madole) on infant categorization using a similar temporal touching coding

- Line 98 – “novel noun generalization” is used in the literature to refer both to novel words with familiar referents (as is the case here), but also novel words with novel referents (such as the shape bias work). Distinguishing which the authors mean is critical.

- Line 150 – “that the distractors are a priori more salient (see above) than the taxonomic choice” – it is unclear what “above” refers to or why they are more salient (because prior lit supposes them to be, or that they were experimentally manipulated to be)

- Line 331 – it is not clear what this warm-up phase would look like or entail (or why it is helpful) if there is nothing on the screen, but the primary task is behavior (looking) toward the screen.

- Line 376 – “significantly different” – in what direction? Which group was faster?

- Line 410-411 – “can be interpreted as an attempt to find a relation between X and Y” – is this universally true and can we be sure of it? Children may be switching gaze not for comparison/relation purposes but because of habituation and boredom with the prior item, e.g. This should be more carefully phrased (as should similar places in the paper).

- Please give specific p-values unless it is <.001

- It might be useful to have a timecourse figure showing moment-to-moment (usually 10ms bins) gaze patterns to each item. Such a figure would illuminate both the early-mid-late patterns as well as switches in relation to time.

- Line 476-477 – something is off grammatically here (I think there is a repeated “was an interaction” phrase)

- The main aims listed in the first paragraph of the discussion do not seem to be identical to those listed at the end of the intro. I suggest listing the aims explicitly in both places, using parallel wording and copying that wording throughout the results as well.

Reviewer #2: This manuscript presents a unique study investigating the real time processes associated with decision making in a noun generalization task. Five-, 6-, and 8-year-old children completed a comparison task, wherein they saw five items. Two items, the learning items, were pictured at the top of the screen and were identified using a shared novel label; the remaining three items, the generalization items, were pictured at the bottom of the screen. Participants were tasked with identifying which generalization item best corresponded with the learning items. In particular, participants could choose an item that was taxonomically similar (ultimately the “correct” choice), perceptually similar, or a unique distractor. In addition to the choice children made, the authors’ novel contribution was analyzing children’s eye movements during the task as a means of identifying the strategies children may use to make their decisions. This measure revealed that children likely use an elimination strategy, at first actively comparing the learning items for shared similarity before systematically comparing generalization targets to arrive at their decision. Additionally, there seemed to be a counterintuitive progression with looking behavior, where 5-year-old children made fewer switches compared to older children, possibly indicating fatigue or early decision-making.

Overall, I appreciate the novel addition of eye-tracking to the study of noun generalization and how the authors thoughtfully frame this study in light of the body of work on comparison in category development. However, there are a few outstanding issues that if addressed would greatly strengthen the manuscript.

First, there were many places in the introduction where a clear definition of the task would have been helpful. The authors frequently refer to a “comparison task” and will reference an “analogical task” but do not take the time to adequately summarize either methodology. Given that the present study builds on this prior work, I think offering a clear overview of the critical methods and then using consistent terminology to refer to the tasks would greatly help the reader better understand the greater context of the paper.

Second, the introduction is heavily focused on the role of alignment in perceptual and conceptual similarity, but it pays little attention to the body of work on noun generalization in early childhood. In particular, I was a bit surprised that there was only a brief mentioning of the Novel Noun Generalization task (and only two references on it), especially since readers of the journal may look at the title of this manuscript and similarly be surprised that that this body of work was not addressed more fully. Here, I think an important contribution of the work shows patterns of noun generalization later in childhood, but the impact is only made once the reader is aware that most previous research on NNG is earlier in childhood. Moreover, I was really surprised that the authors did not have a language measure as a control variable in their models; given the abundant literature showing the role of vocabulary in noun generalization, I think its absence in the present study is worth discussing at some point in either the method or general discussion.

Third, the authors conclude that the results are consistent with “alignment” but never really offer an alternative theoretical perspective, either in the introduction or discussion. If children are not visually aligning and comparing the stimuli, what else would they be doing? Clarifying the alternative option would strengthen the conclusions drawn, especially in lines 607-608. Moreover, I wonder if including a trial-by-trial analysis would provide insight into the different strategies children use. Here, the "first gaze" information is useful, but I could imagine that as children gain familiarity with the task that their first gazes and strategies may change.

Fourth, though this may be a “minor comment”, I was surprised that the authors did not include the rationale for the ages, especially since 7-year-olds were not included altogether. Given the interesting developmental progression the authors discuss, having more context for why participants were selected would be helpful, especially since there appears to be a big transition happening with respect to looking behavior and taxonomic decision-making during this developmental period.

Finally, I struggled at times with the overall readability of the paper. The structure of the paragraphs could be hard to follow, especially when laying out the different strategies, discussing the similarity ratings for the stimuli, and reporting the results. Though the approach to the study was novel and interesting to learn about, I think working carefully with a copy editor (in addition to some of the suggestions I make below) would greatly strengthen the readability of the manuscript overall.

Additional Comments:

Line 19 – Abrupt start to the abstract. I think it may help the audience to describe the comparison task before discussing results and limitations.

--Paragraph starting with line 54 – it would help to name the paradigm. It is a “Novel Noun Generalization” task.

--Line 59: Not sure what the authors mean by “learning designs”

--Line 61: for “learning stimuli”, do the authors mean two or more objects belonging to the same basic category? I wonder if an example would clarify what they mean, as the sentence is hard to understand.

--Line 64: Genter and Namy’s (1999)

--Line 65: I suggest replacing “imaginary” with “novel”

--Line 69: The authors mention a “learning phase” but do not explain what happens during the phase; I think clarifying this point would also help distinguish what Gentner and Namy found in the no-comparison condition

--Lines 79 and 80: eliminate “would”

--Line 80: replace “neglect” with “ignore”

--Line 81: “authors” is ambiguous

--Line 81-82: Getting a little lost on the different paradigms. I think a clear description of each would be really helpful, especially as the authors continue to refer to them.

--Line 89: Unsure what the authors mean by “mapping” -> what are individuals mapping? Labels?

--Line 98: remove “the”

--Line 109: Hard to follow; again, I think a clear definition of the terminology would be very helpful

--Line 114-115: Consider defining “source domain” and “generalization-target domain”. For this paragraph, I would consider adding a figure to depict the different combinations of strategies

--128-131 -> hard to follow. I wonder if a figure for previous paragraph would also facilitate discussion of strategies in this paragraph

--134 -> Not sure I understand how the Task goal strategy differs from previous strategies. Are these all strategies used within the same paradigm? Is a “task goal” strategy something that can be addressed by trial-by-trial analyses of the task?

--177: Wonder if a discussion of semantic distance earlier in introduction is warranted to provide motivation for this manipulation

--249: replace “&” with “and”; remove comma after “Witt”

--265: “Students” = what age? Who normed them? Overall, I wonder if a description of the similarity ratings is best saved for the supplemental materials. At present, it distracts from the focus of the paper.

--319: replace common with semi-colon

--330: move period inside quotation marks

--331: did not understand what happened during the warm-up trial

--341: Replace “randomly assigned” with “randomized”

--344: Did they prescreen children’s knowledge of items? What was “very low”? Were trials dropped if children did not know the items?

--364-5: Report t-values

--393: “was” with “were”

--476-477: repetition of “interaction”

--502: The work on the first gaze makes me wonder if first gaze aligns with hearing the label and if first gaze changed across trials as children became more familiar with the task

--548: Achievers -> need to define this sooner instead of “see below”; is there precedent for this approach?

--609 – fixation (remove “s”)

--610 – generalization options? I think consistent labeling would be very helpful

--615-20 – a bit hard to follow

--623 = is -> are

--626 = remove “which is the pattern we found” as it is redundant

--Paragraph 621 = watch for redundancy

--637 = I found myself asking “how do you know?”

--641 = “ration” => “ratio”

--659-661 = hard to follow

--664 = not sure what the authors mean by control

--704 = “problems” -> “problem”

6. PLOS authors have the option to publish the peer review history of their article (what does this mean?). If published, this will include your full peer review and any attached files.

Reviewer #1: No

Reviewer #2: No

---

## [Author Response · Author response to Decision Letter 0]

13 Sep 2023

Dear Reviewers, 

Thank you both for all your insightful comments. We have thoroughly revised our manuscript in light of all your remarques that have greatly helped to improve it. We would like to specifically thank Reviewer 1 for your discussion on the importance of the perceptual distractor and its implication on the hypothesizes made about generalization strategies. We have completely added this aspect to the manuscript that is consequently a much stronger piece of work. We would also like to thank Reviewer 2 for your extremely helpful comments for improving the manuscript's clarity for all readers especially those not necessarily familiar with our domain. We have added sections in the introduction, clarified the terminology and hope that the manuscript is now much more accessible to all readers.

---

## [Decision Letter · Decision Letter 1]

20 Oct 2023

PONE-D-22-35572R1How children generalize novel nouns : an eye tracking analysis of their generalization strategies.PLOS ONE

Dear Dr. Stansbury,

Thank you for submitting your manuscript to PLOS ONE. After careful consideration, we feel that it has merit but does not fully meet PLOS ONE’s publication criteria as it currently stands. Therefore, we invite you to submit a revised version of the manuscript that addresses the points raised during the review process.

We look forward to receiving your revised manuscript.

Kind regards,

Antoine Coutrot

Academic Editor

PLOS ONE

Journal Requirements:

Reviewers' comments:

Reviewer's Responses to Questions

**Comments to the Author**

1. If the authors have adequately addressed your comments raised in a previous round of review and you feel that this manuscript is now acceptable for publication, you may indicate that here to bypass the “Comments to the Author” section, enter your conflict of interest statement in the “Confidential to Editor” section, and submit your "Accept" recommendation.

Reviewer #1: (No Response)

Reviewer #2: (No Response)

2. Is the manuscript technically sound, and do the data support the conclusions?

Reviewer #1: Partly

Reviewer #2: Partly

3. Has the statistical analysis been performed appropriately and rigorously? 

Reviewer #1: No

Reviewer #2: Yes

4. Have the authors made all data underlying the findings in their manuscript fully available?

Reviewer #1: Yes

Reviewer #2: Yes

5. Is the manuscript presented in an intelligible fashion and written in standard English?

Reviewer #1: Yes

Reviewer #2: No

6. Review Comments to the Author

Reviewer #1: The current manuscript is a revision of one I have reviewed previously. The authors have taken care to significantly improve the introduction and discussion of background literature, expanding the scope of the theories and, critically, identifying the key point of the current study (page 4, line 70). There remains some points of clarity, however, as detailed below.

The discussion of analogical reasoning and parallels with that task are less clear – while both analogical reasoning and noun generalization tasks require comparison and generalization, the analogical reasoning task by design is a much more cognitive complex task and requires a much higher level of logistical processing. The “A is to B as C is to D” comparison is very difficult for young children to do, meaning the ages the tasks are used with also vary. By trying to draw analogical reasoning literature into the current study just makes the current study hypotheses more convoluted and posits processes that don’t need to be proposed in order to examine looking behavior in generalization tasks. Similarly, by line 267, there are hypotheses being proposed that directly tie to the strategies, but since the task here is not an analogical reasoning task, it is unclear how those might apply or if they would even be relevant.

There are some remaining questions and possible concerns about decisions in the methods and analyses as well. First, on Line 330 – choosing a sample size simply because a prior study did is not a valid rationale for power. While the authors may not have conducted an a prior power analysis to determine the sample size, post hoc inclusion of power that confirm that this sample is sufficient to detect the effects anticipated is needed. This is especially true because of the large models needed to analyze the looking behavior. Second, line 545 indicates that post-hoc, a key variable was removed from the analysis because it wasn’t significant. Yet learning distance seemed to be a critical element of the design and hypotheses, thus removing it simply to “improve robustness” of the results sounds an awful lot like p-hacking. I understand if there is a better model fit without it, but then more details must then be presented demonstrating that patterns of significance do not chance and the full planned analyses need to be presented at least in the supplement. Another concern about the data is that 19 children were dropped for not having good calibration and additional 32 for poor tracking. While the sample size is large, data needs to be presented that this significant drop in participants did not impact the overall power

Minor points:

- Citation 21 listed on line 27 and 78 doesn’t seem to match the argument. From what I can tell, citation 21 is not about the shape bias..

- Line 88 – the taxonomic choice won’t always be the “correct” item – which is correct depends on the task, objects available, semantic context, etc. Either correct this sentence to specify that in this specific example in figure A, taxonomic is most likely correct, or better yet, just remove the word “correct”. A similar issue to this comes up on page 10 (lines 219) when the authors presume that the taxonomic is the correct answer. While the authors may hypothesize that taxonomic may be the more frequent response and they may analyze “proportion of choices to taxonomic choice”, assuming that it is the “correct” answer for a child is overstepping what we can know about children a priori. Being more careful with the wording used is important here.

- Line 229 – need a citation for the fact that 5-year olds are distracted by salience (and that it is actually distraction vs. immature categorization abilities)

- Line 331 – define “some”

- Line 346 – which “learning items” are you referring to here? It would seem as though both the curb chain and the watch are both perceptually and conceptually similar to the bracelet, but the curb chain is much more perceptually similar and watch a bit less so.

- Please give specific p-values when available

- The high vs. low achieving analyses need to be more clearly outlined as exploratory as there was no a priori plan for this nor hypotheses until after the data were collected.

Reviewer #2: I would like to thank the authors for their careful consideration of my original feedback. Their revision provided much broader theoretical insight, and the clearer description of the tasks used made the study as a whole much easier to understand. Overall, I think this manuscript benefitted tremendously from the revision.

This said, I have a few remaining points of concern that I think should be addressed to strengthen the manuscript before being published.

First, I noticed that the authors elaborated more fully on the rationale for 2D as opposed to 3D stimuli (line 362) in the revised manuscript, but the rationale presented seemed questionable. In particular, many iterations of the NNG task with toddlers have been done with 3D objects (e.g., Landau et al., 1998; Samuelson & Smith, 2002; Perry et al., 2010), and there has been a surge in the use of head-mounted eye-trackers to better understand the real-time visual dynamics of children's interactions with objects, used with children as young as 18 months of age (e.g., Yoshida & Smith, 2008; Yu & Smith, 2012). I think it would be beneficial to further clarify why 2D objects were a particular strength (or necessity) of this study's design.

Second, relatedly, I think there needs to be further clarification on the use of familiar objects. Although the authors mention that these categories have been used in similar studies before, I think the conclusions drawn from the study's results may be significantly impacted by the fact that children already have labels for the objects and categories. Additional rationale and/or discussion on the use of familiar as opposed to novel stimuli (as is seen in many other word learning studies) and how it may have impacted the results should be included in the manuscript.

Third, it would be beneficial to more clearly define the different time bins. In a recently published study, Bakopoulou et al. (2023) illustrated that the presentation of the novel noun shifted children's attention in a novel noun generalization task, especially for children who displayed a strong shape bias. Knowing that the timing of the prompt impacted the cascade of looks in a younger age group, it seems plausible that how the bins were defined could significantly impact the interpretation of the results in the present study.

Finally, I found myself questioning why the authors would restrict their analyses to trials with only a "correct" taxonomic match. In particular, as I read lines 632-633, I was not surprised by the results because we know that individuals are likely going to turn their attention to the object they plan to select so as to guide their reach (or point, in this case). It made me wonder if the pattern of looking behavior was different for trials with errors, perhaps especially for the low generalizers. Moreover, though the authors shared in their response that they were interested in average looking behaviors across the 14 trials, I was still left wondering if overall attention to the distractor items decreased across trials, especially for children who were good at generalization.

In addition to these larger points, here are other minor points of feedback:

--The revised abstract could be hard to understand at times. In particular, I think it would be beneficial to more clearly define what is meant by learning items and to more clearly delineate between the different generalization options. In it's current form, the organization of the options is quite hard to track; making these examples clearer is critical, especially given the framing of the study.

--line 19: favor (not favors)

--line 31: I recommend using a semi-colon instead of "and"

--line 97: Should revise to say "single-learning item"; the current phrasing suggests that participants see only the learning item before the test items (which would heavily demand working memory), whereas most often the learning and generalization items are presented simultaneously. The novel aspect of this study is to have multiple learning items, so revising the language here will be helpful.

--line 159: remove the comma before the question mark

--line 170: I think removing "and they have been described in previous studies" would improve the clarity of the writing

--line 189: "has" instead of "have"

--line 209: hard to follow; consider revising, perhaps to something more in line with discussing the real-time developmental differences in processing during the task

--line 235: I appreciate that the authors added a rationale for the ages, but I struggled to understand the rationale for 5-year-olds here, especially knowing that eye-tracking studies have been conducted with much younger children.

--overall, the use of the word "aims" appears frequently in the introduction. Given the 5 aims that appear in the intro and discussion, the authors should examine the other places where they use the word "aim" to see if a different word would be more appropriate

--line 395: generalization item"s"

--line 395-402: I had a hard time interpreting the results as presented here.

--all p-values should have the exact value reported unless less than .01

--line 496: Did the authors mean to include a main effect of age as well? The statistics are reported for it in line 497.

--lines 508-512: I appreciate the detailed reporting of the t-tests but recommend revisiting the prose, as organization here made results a little hard to interpret.

--line 543: effect (not size effect)

--line 546: double negative hard to follow; perhaps instead say that since generalization distance interacted significantly with performance, it was included in the remaining analyses

--line 584: This paragraph should be moved up to earlier in the results section so as to clarify the corrections used throughout the analyses

--lines 590-591: slightly repetitive

--line 593: Because our sample size was large, we…

--line 612: analyses

--lines 613-614: need to clarify the contrasts of interest

--lines 685 & 686: comma after end

--line 896: seems like a word is missing

7. PLOS authors have the option to publish the peer review history of their article (what does this mean?). If published, this will include your full peer review and any attached files.

Reviewer #1: No

Reviewer #2: No

---

## [Author Response · Author response to Decision Letter 1]

4 Dec 2023

Thank you for your reply to our manuscript and for your comments. Overall, we feel you very rightfully highlighted that the manuscript would benefit from a more detailed report of the methodological decisions taken throughout the study. We feel that addressing these aspects has helped strengthen the manuscript and make it more accessible for future research. We have addressed all of the issues raised by the two reviewers and hope that the manuscript now meets PLOS ONE’s publication criteria. 

We have carfully taken all comments into consideration and have attached a full response to these in a letter with the submission.

---

## [Editor Report · Decision Letter 2]

20 Dec 2023

How children generalize novel nouns : an eye tracking analysis of their generalization strategies.

PONE-D-22-35572R2

Dear Dr. Stansbury,

We’re pleased to inform you that your manuscript has been judged scientifically suitable for publication and will be formally accepted for publication once it meets all outstanding technical requirements.

Kind regards,

Antoine Coutrot

Academic Editor

PLOS ONE
---

## [Editor Report · Acceptance letter]

25 Mar 2024

PONE-D-22-35572R2 

PLOS ONE

Dear Dr. Stansbury, 

I'm pleased to inform you that your manuscript has been deemed suitable for publication in PLOS ONE. Congratulations! Your manuscript is now being handed over to our production team.

Kind regards, 

on behalf of

Dr. Antoine Coutrot 

Academic Editor

PLOS ONE